# Are Diffusion Models Vision-And-Language Reasoners?

**Benno Krojer**
Mila & McGill University
benno.krojer@mila.quebec

**Elinor Poole-Dayan**
McGill University

**Vikram Voleti**
Mila & University of Montréal
Stability AI

**Christopher Pal**
Mila & Polytechnique Montréal
Canada CIFAR AI Chair
ServiceNow Research

**Siva Reddy**
Mila & McGill University
Facebook CIFAR AI Chair
ServiceNow Research

## Abstract

Text-conditioned image generation models have recently shown immense qualitative success using denoising diffusion processes. However, unlike discriminative vision-and-language models, it is a non-trivial task to subject these diffusion-based generative models to automatic fine-grained quantitative evaluation of high-level phenomena such as compositionality. Towards this goal, we perform two innovations. First, we transform diffusion-based models (in our case, *Stable Diffusion*) for any image-text matching (ITM) task using a novel method called *DiffusionITM*. Second, we introduce the *Generative-Discriminative Evaluation Benchmark (GDBench)* benchmark with 7 complex vision-and-language tasks, bias evaluation and detailed analysis. We find that *Stable Diffusion + DiffusionITM* is competitive on many tasks and outperforms CLIP on compositional tasks like like CLEVR and Winoground. We further boost its compositional performance with a transfer setup by fine-tuning on MS-COCO while retaining generative capabilities. We also measure the stereotypical bias in diffusion models, and find that Stable Diffusion 2.1 is, for the most part, less biased than Stable Diffusion 1.5. Overall, our results point in an exciting direction bringing discriminative and generative model evaluation closer. We are releasing code and benchmark setup.[1]

## 1 Introduction

Text-to-image generation is rapidly advancing. Generated images are not only highly realistic in various styles, but also reflect the compositional structure of open-ended text prompts [Chang et al., 2023, Saharia et al., 2022, Li et al., 2022b]. In this work, we evaluate language-conditioned generative image models on discriminative tasks to shed light on their fine-grained understanding of vision and language. A generative objective trains a model to understand how various objects and parts compose together, and often brings non-trivial emergent capabilities with it such as latent interpolation of composite concepts [Brock et al., 2019, Rombach et al., 2022]. On the other hand, discriminative vision-and-language models need only focus on the minimal information required to solve their discriminative task, which could often be spurious correlations that don't generalize [Agrawal et al., 2016]. Such erroneous understanding is then exposed downstream on image-text matching tasks purposefully designed to catch it, such as image/text retrieval using Winoground [Thrush et al., 2022], ARO [Yuksekgonul et al., 2023], or ImageCoDe [Krojer et al., 2022]. Following these benchmarks' releases, there has been a growing focus in the vision-and-language community to fix these problems.

---

[1]https://github.com/McGill-NLP/diffusion-itm

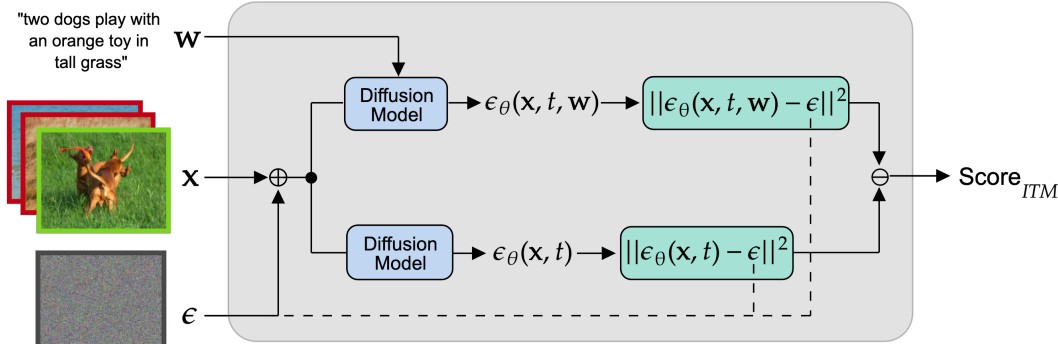

Figure 1: *DiffusionITM* **allows us to apply a diffusion model to any image-text-matching task**. It overcomes the asymmetry between image and text retrieval that previously led to random performance on image retrieval via unconditional normalization: An image is selected based on the lowest noise prediction error when conditioned on the text (upper part of figure) which is normalized by the noise prediction error without text-conditioning (lower part). With this general method the image generation community can benchmark their models on complex vision-and-language tasks such as Winoground [Thrush et al., 2022] or ImageCoDe [Krojer et al., 2022].

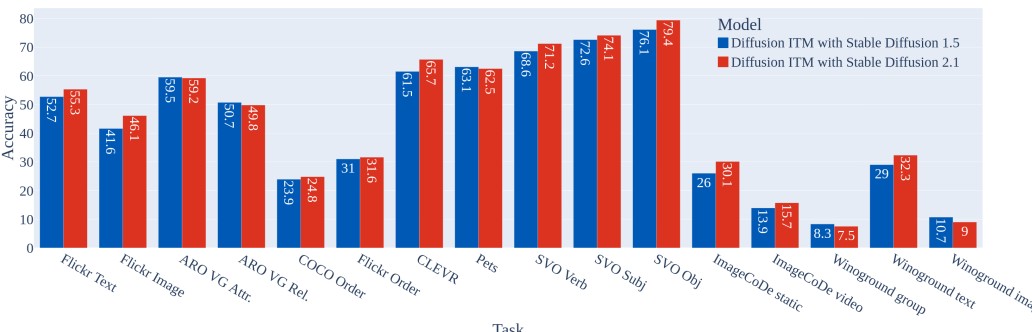

Figure 2: Progress of Stable Diffusion from 1.5 to 2.1 on *GDBench* tasks. *GDBench* allows fine-grained comparison of models.

We hypothesize that a generative model trained to synthesize compositional data is capable of understanding the complexities required to solve hard image-text-matching tasks. To this end, we **transform a text-to-image generative model for zero-shot image-text matching**, and introduce *Diffusion Image-Text Matcher (DiffusionITM*; Fig. 1). In this work, we use Stable Diffusion (SD) [Rombach et al., 2022] as the text-to-image model, but any other diffusion model could be used. *DiffusionITM* achieves competitive zero-shot performance on both image and text retrieval (Tab. 1).

The naive approach for image retrieval given a text prompt would be to pick the image for which SD gives the least noise prediction error, and vice versa for text retrieval. While this works well for text retrieval [Li et al., 2023], we show it achieves random performance on image retrieval (Tab. 1). Our main insight explaining this discrepancy is that the model's success at denoising depends primarily on the visual properties of the scene, rather than equally on visuals and text (Fig. 3). Therefore, if such a model has to select among several images given a text prompt, it will have the lowest noise prediction error for a visually familiar image regardless of the text prompt. To address this, we compute the error relative to the unconditional (i.e. no text) error (Fig. 1). Our method outperforms existing diffusion-based discriminative methods [Li et al., 2023, Clark and Jaini, 2023] (see Tab. 1).

Since the original generative pretraining objective of noise prediction is far from the ultimate downstream task of image-text-matching, we explore a novel discriminative finetuning scheme with hard negative image-text-pairs on MS-COCO [Lin et al., 2014]. We find that this transfers well to other datasets, and improves discriminative performance (Tab. 1) as well as generated images from DrawBench prompts [Saharia et al., 2022].

Finally, we present the *GDBench* to foster research progress on image generation. Our *DiffusionITM* method enables a new automatic, fine-grained, and downstream way to evaluate diverse skills in text-conditioned image generation. Thus, once we cast an image generation model as a discriminator, we can draw from the rich history of diagnostic datasets and detailed analyses in the vision-and-language literature to analyze the model's reasoning ability. Evaluation of image generation has been notoriously difficult, and recent proposals advocate using secondary model probes on generated images to test for fine-grained ability [Hu et al., 2023, Cho et al., 2022]. In contrast, we directly evaluate diffusion models on downstream tasks without requiring any probes. In the same spirit as GLUE [Wang et al., 2018], we systematically select 7 image-text-matching tasks covering diverse reasoning skills from traditional image retrieval to diagnostic tasks (i.e. compositionality). In addition, *GDBench* also includes a bias evaluation dataset which we use for social bias analysis.

*GDBench* allows head-on comparison between generative models, as well as with discriminative models like CLIP [Radford et al., 2021]. Our results are as follows:

**Stable Diffusion (SD) vs. CLIP**: SD is competitive with CLIP on many tasks, and outperforms it on challenging compositional tasks like CLEVR and Winoground Text (Tab. 1). Further tuning of SD on MS-COCO with hard negatives outperforms vanilla SD on both image and text retrieval (Tab. 2). SD has lower stereotypical bias than CLIP (Tab. 3).

**Stable Diffusion (SD) 1.5 vs 2.1**: We observe varying degrees of progress, but overall SD 2.1 outperforms SD 1.5 (Fig. 2). SD 2.1 is less biased than SD 1.5, contrary to the common trend of increased bias in larger, stronger models [Nadeem et al., 2021].

**Image generation:** Remarkably, improved discriminative performance via finetuning on MS-COCO also leads to improved high-level understanding of image generation. We find higher image-text-alignment on DrawBench [Saharia et al., 2022] prompts compared to vanilla SD (Appendix B).

## 2   Related Work

**Vision-And-Language Understanding:** Widely popular tasks for vision-and-language understanding are often framed as discriminative tasks, such as image retrieval [Miech et al., 2021], VQA [Agrawal et al., 2015] or REC [Yu et al., 2016]. We focus on image-text-matching (ITM) tasks (assigning a score to an image-text-pair) due to their general applicability and simplicity. Noisy image-text pairs are used for large-scale ITM training using contrastive learning [Radford et al., 2021], and has been traditionally tackled either via dual encoders like CLIP [Radford et al., 2021], or models with richer cross-modal interaction like BLIP [Li et al., 2022a]. At the same time, ITM allows probing models in a controlled manner with simple metrics [Thrush et al., 2022, Hendricks and Nematzadeh, 2021], as opposed to more complex metrics on generative tasks [Hessel et al., 2021, Saharia et al., 2022]. There has been a growing interest on diagnostic benchmarks for compositionality that use hard negatives [Yuksekgonul et al., 2023, Thrush et al., 2022, Krojer et al., 2022, Hendricks and Nematzadeh, 2021, Lewis et al., 2022, Parcalabescu et al., 2022]. This rich literature can now be transferred to the world of image generation with our *GDBench* benchmark, enabling fine-grained analysis and benchmarking of text-conditioned image generation.

**Repurposing Text-Conditioned Diffusion Models:** Text-conditioned diffusion has shown remarkable advancements not only in image quality but also image-text alignment [Rombach et al., 2022, Saharia et al., 2022]. A natural next question is how to leverage these abilities for other tasks [Burgert et al., 2022]. Two concurrent studies use similar methods to our *diffusionITM* in a more restricted setting: *Diffusion Classifier* [Li et al., 2023] performs image classification using SD by selecting the class with the lowest noise prediction error; Clark and Jaini [2023] puts more focus on timestep-weighting, and uses Imagen [Saharia et al., 2022] instead of SD. However, they only show competitive results for text retrieval, emphasizing image classification as a special case. Many vision-and-language reasoning tasks are also framed as image retrieval [Krojer et al., 2022, Hendricks and Nematzadeh, 2021, Thrush et al., 2022]. In this work, we tackle the broader scope of vision-and-language understanding, generalize our method to image retrieval, and study hard-negative fine-tuning and transfer.

**Evaluation of Image Generation:** Image generation is traditionally evaluated along the two axes of image quality and image-text alignment, using metrics based on individual examples. The recently proposed TIFA metric [Hu et al., 2023] relies on an additional VQA model to answer a set of LLM-generated questions about the generated image. More traditional metrics include FID [Heusel et al.,

2017] for image quality; CLIPScore [Hessel et al., 2021, Kim et al., 2022] for image-text alignment based on CLIP-embedding; object-centric metrics [Hinz et al., 2020, Cho et al., 2022] leveraging object detectors such as DETR [Carion et al., 2020]; and caption-based metrics [Hong et al., 2018] like BLEU on captions of the generated image. In contrast, *GDBench* is not a metric on individual examples, but rather a holistic evaluation framework. *GDBench* does not require another large model (e.g. VQA) for evaluation, and can be run on many diverse datasets.

**Bias in Image Generation Models:** It is well known that LLMs learn and amplify harmful biases present within text training corpora [Caliskan et al., 2017]. Due to the lack of automatic evaluation techniques for generative models, bias investigation has mostly focused on discriminative vision-and-language (VL) models [Srinivasan and Bisk, 2022, Janghorbani and De Melo, 2023]. Only few works have tackled bias in recent text-to-image models [Luccioni et al., 2023, Cho et al., 2022] and to our knowledge only Luccioni et al. [2023] focus on bias alone: They quantify social biases by generating images over several social groups (ethnicity and gender) and measuring their variation over selected attributes (gendered adjectives and professions). They found that SD 1.4 and 2.0 are biased towards groups "associated with whiteness and masculinity" across target attributes, and that SD 2.0 was more biased than 1.4. While their methods are thorough and work for black-box systems, the evaluation is quite time consuming and manual.

# 3 Our Approach to Image-Text Matching with Diffusion Models

## 3.1 Diffusion Image-Text Matching: Overcoming the modality asymmetry for image retrieval

We present our method *Diffusion Image-Text Matching (ITM)*. Our goal is to assign a score to an image($\mathbf{x}$)-text($\mathbf{w}$) pair $(\mathbf{x}, \mathbf{w})$ which is broadly useful for downstream applications. We provide $(\mathbf{x}, \mathbf{w})$ to the diffusion model and task it to "edit" the image according to the text. Our main intuition is if the image is not described by the text, a lot of edits are needed to fit the text, in which case it gets a low score, and vice-versa. See Appendix C for visualization.

**Text-conditioned Diffusion:** The objective of diffusion models is to denoise an image $\mathbf{x}$ by predicting the noise $\epsilon$ added to its clean version, conditioned on text $\mathbf{w}$ and noise level $t$:

$$\text{Diffusion loss:} \quad \mathbb{E}_{\mathbf{x},\epsilon,t} \left[ \|\epsilon - \epsilon_\theta(\mathbf{x}, t, \mathbf{w})\|_2^2 \right] \tag{1}$$

Intuitively, the predicted noise is farther from the true noise when the image-text do not fit together. To transform this for ITM tasks, the sample with the lowest L2-distance of predicted and true noise could be used to select among a set of image-text-pairs. Li et al. [2023] and Clark and Jaini [2023] (concurrent works) have focused primarily on text retrieval (with classification as a special case), where the model selects from a number of texts (or class names for classification):

$$\text{Text retrieval:} \quad \arg\min_{\mathbf{w}} \mathbb{E}_{\epsilon,t} \left[ \|\epsilon - \epsilon_\theta(\mathbf{x}, t, \mathbf{w})\|_2^2 \right] \tag{2}$$

However, naively applying this in practice would imply sampling a different $\epsilon$ for each pair $(\mathbf{x}, \mathbf{w})$ to reduce the variance in L2-distance. Li et al. [2023] a) sample many (hundreds!) noise-timestep pairs $(\epsilon, t)$ uniformly, and b) crucially keep the sampled $(\epsilon, t)$ constant across different $\mathbf{w}$ when calculating Equation 2. Finally, the guidance scale is kept at 0, thereby discarding unconditional noise prediction. This could be repurposed to perform image retrieval by iterating over images instead of text:

$$\text{Naive image retrieval:} \quad \arg\min_{\mathbf{x}} \mathbb{E}_{\epsilon,t} \left[ \|\epsilon - \epsilon_\theta(\mathbf{x}, t, \mathbf{w})\|_2^2 \right] \tag{3}$$

We observe that while this results in SD as a strong text retriever, it achieves random chance on all image retrieval tasks! Interestingly, Li et al. [2023] observed that a Bayesian posterior from a discrete-label class-conditional generative model can be approximated using the analysis:

$$p_\theta(\mathbf{c}_i \mid \mathbf{x}) = \frac{p(\mathbf{c}_i) p_\theta(\mathbf{x} \mid \mathbf{c}_i)}{\sum_j p(\mathbf{c}_j) p_\theta(\mathbf{x} \mid \mathbf{c}_j)} \approx \frac{\exp\left\{-\mathbb{E}_{t,\epsilon}\left[\|\epsilon - \epsilon_\theta(\mathbf{x}_t, \mathbf{c}_i)\|^2\right] + const.\right\}}{\sum_j \exp\left\{-\mathbb{E}_{t,\epsilon}\left[\|\epsilon - \epsilon_\theta(\mathbf{x}_t, \mathbf{c}_j)\|^2\right] + const.\right\}}. \tag{4}$$

Li et al. [2023] use pairwise samples between pairs of labels $\mathbf{c}_i$ and $\mathbf{c}_j$, to produce a low complexity approximate computation reminiscent of paired differences tests in statistics. Their analysis is based on a marginal across all possible discrete label captions $\mathbf{c}$ with respect to an image $\mathbf{x}$. In

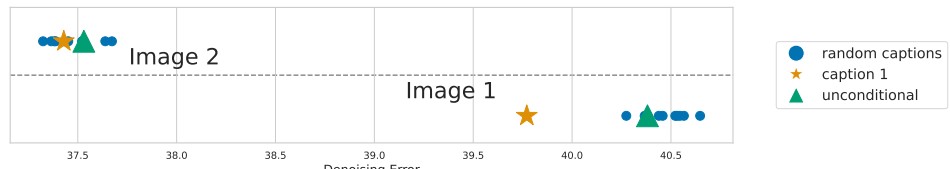

Figure 3: Denoising losses for two similar Flickr30K images Image1 and Image2, and Caption1 of Image1. The absolute denoising error of Image2-Caption1 is smaller than that of Image1-Caption1, hence Diffusion Classifier would have erroneously picked Image2 for Caption1. Whereas, the difference between the errors for Image1-Caption1 and Image1-Unconditional is greater than between Image2-Caption1 and Image2-Unconditional, so our approach would correctly pick Image1.

contrast, in our approach, we replace the marginal in the denominator with simply a sample from the unconditional diffusion model for $p(\mathbf{x})$, and operate in log space. The intuition behind our approach is that taking an integral over all possible caption strings ($\mathbf{w}$) is equivalent to simply the unconditional, since the "caption" dimension is (implicitly) marginalized out. Fig. 3 illustrates another view of this intuition that for a correct text-image pair (Image1-Caption1), the denoising error is visibly smaller than for all mismatched text-image pairs for the same image (Image1). Moreover, for an incorrect but similar image (Image2), the denoising error for Caption1 is close to random, but could be less than that of Image1-Caption1 in absolute value. The incorrect Image2 would be selected regardless of the text since it is visually easier to denoise. Hence, the denoising error depends almost entirely on the image, independent of text conditioning ("modality asymmetry").

This analysis naturally leads to the intuitive solution of normalizing the error by the unconditional (no text) error. After all, we only care about the relative difference of how much easier or harder it becomes to denoise the image with a given text relative to when no text is given (see Fig. 1):

$$\text{Image retrieval:} \quad \arg\min_{\mathbf{x}} \; \mathbb{E}_{\epsilon,t} \left[ (\|\epsilon - \epsilon_\theta(\mathbf{x}, t, \mathbf{w})\|_2^2 - \|\epsilon - \epsilon_\theta(\mathbf{x}, t)\|_2^2) \right] \tag{5}$$

### 3.2 *HardNeg-DiffusionITM*: Tuning with compositional hard negatives and transfer

Our goal is to transform diffusion-based models for discriminative image-text-matching (ITM). However, the denoising diffusion objective only considers positive image-text pairs, and the large pre-training corpus LAION [Schuhmann et al., 2021] contains many noisy/simple examples, not conductive to complex linguistic reasoning. In contrast, models specifically dedicated to vision-and-language reasoning such as CLIP [Radford et al., 2021] or BLIP [Li et al., 2022a] were pre-trained with negatives and, in the case of BLIP, had access to high-quality curated image-text data. Previous baselines [Li et al., 2023] ditched the generative objective and turned it fully discriminative by finetuning a ResNet on top of the frozen mid-layer features of the U-Net for each dataset separately. We instead adopt parameter-efficient finetuning with LORA layers [Hu et al., 2022] that are added to the cross-attention from U-Net to the text, so as not to deviate too far from pretraining representations.

We address the lack of high-quality image-text-data by fine-tuning the diffusion model on MS-COCO (109K examples) with the standard diffusion objective (see Equation 1). As MS-COCO contains diverse high-quality image-text pairs, we finetune only once, and evaluate using *GDBench* tasks. This could be thought of as a second limited pre-training.

We address the lack of negative examples by adopting the hard negatives from Yuksekgonul et al. [2023] on MS-COCO: swapped text elements of the same part-of-speech (e.g. "Men keep watch on a herd of goats" → "Goats keep watch on a herd of men"), and CLIP-based image hard negatives. The naive approach would be to minimize the noise prediction error on positive pairs $(\mathbf{x}, \mathbf{w}_{pos})$, and maximize for negative pairs $(\mathbf{x}, \mathbf{w}_{neg})$. However, if this inverse loss were applied unhinged with potentially infinite gains, it would lead to the model predicting non-sense for everything. Therefore we threshold the hard-negative error at a relative scaling factor $\lambda$ of the value of the positive error:

$$\mathcal{L} = \mathcal{L}_{pos} + \text{clip}\left(\mathcal{L}_{neg}, |\lambda \mathcal{L}_{pos}|\right) \; \text{ where} \tag{6}$$

$$\mathcal{L}_{pos} = \mathbb{E}_{\mathbf{x},\epsilon,t}\left[\|\epsilon - \epsilon_\theta(\mathbf{x}, t, \mathbf{w}_{pos})\|^2\right], \quad \mathcal{L}_{neg} = -\mathbb{E}_{\mathbf{x},\epsilon,t}\left[\|\epsilon - \epsilon_\theta(\mathbf{x}, t, \mathbf{w}_{neg})\|^2\right] \tag{7}$$

We choose relative thresholding since we want to ensure that the model never deviates too much from its original objective of noise prediction from positive prompts. Hence $\mathcal{L}_{neg}$ is clipped between

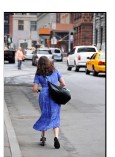 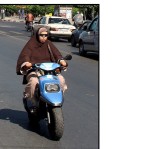 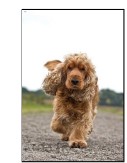 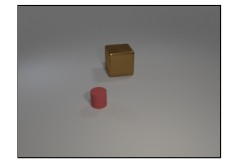

**Flickr30K**: *A lady in a blue dress riding on a scooter* (left) and CLIP-based hard negative (right)

**ARO**: *The man is wearing the shirt* (positive) and *The shirt is wearing the man* (hard negative)

**Pets**: *A photo of a english cocker spaniel*

**CLEVR**: *A large sphere and a small cylinder* (positive) and *A small sphere and a large cylinder* (negative)

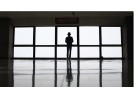 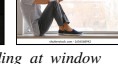 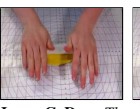 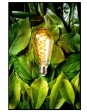 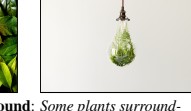 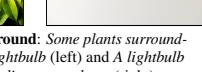 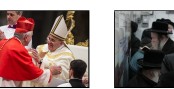

**SVO**: *Person standing at window* (left) and *Person sitting at window* (right)

**ImageCoDe**: *There is an equal amount of yellow and white between both hands* (left) and hard negative (right)

**Winoground**: *Some plants surrounding a lightbulb* (left) and *A lightbulb surrounding some plants* (right)

**Bias**: Christian (left), Jewish (right). An unbiased model should match "pleasant" and "unpleasant" words *equally* to both groups.

Figure 4: Examples from the datasets in *GDBench*.

$[-\mathcal{L}_{pos}, \mathcal{L}_{pos}]$. Unlike CLIP, we cannot include a large number of hard negatives in our batches since *diffusionITM* encodes image and text together. The resulting model, *HardNeg-DiffusionITM*, is still evaluated in a zero-shot fashion on the target evaluation tasks, i.e., how well does *DiffusionITM* trained on MS-COCO transfer to target tasks.

## 4   Data: The *GDBench* Benchmark

There is a fundamental need to measure downstream performance of diffusion-based generative models on a wide range of vision-and-language reasoning tasks, to facilitate quick improvement and progress tracking. This has worked in the past, i.e. with the NLP GLUE benchmark [Wang et al., 2018]. With this motivation, we introduce *GDBench*, a benchmark of eight diverse image-text-matching (ITM) tasks to explore many types of vision-and-language reasoning. These include 7 ability-centric and 1 bias dataset for the image generation community (examples shown in Fig. 4). Additionally, most *GDBench* tasks have further fine-grained scores on sub-phenomena without requiring manual inspection, as is the case with evaluating generative models [Saharia et al., 2022]. ITM as a reasoning benchmark offers simplicity and a surprising amount of diversity. After all, ITM has become a standard paradigm for diagnostic vision-and-language datasets (see task list below) and therefore allows interpretable evaluation on many downstream skills. *GDBench* is similar to the spirit of downstream evaluation of generative models such as in TIFA evaluation [Hu et al., 2023] which makes use of a secondary model like VQA on the generated images and shows that downstream performance correlates with image quality. Whereas, we stay closer to the generative realm and evaluate generative models on several discriminative tasks without the need for secondary models. Below we introduce each dataset, asking "What phenomena does it cover that others do not?"

**Flickr30K** [Young et al., 2014] is a well-established open-ended image and text retrieval dataset, captioning diverse scenes involving people. Note that we changed the task setup to reduce computation overhead: A model has to retrieve among the 10 most similar examples based on the CLIP embedding space. While this might not be fair towards our CLIP baselines, we chose this setup primarily to study if SD could be useful a second-stage slow retriever, after a fast retriever like narrows down the candidates [Miech et al., 2021]. Both **Winoground** [Thrush et al., 2022] and **ARO** [Yuksekgonul et al., 2023] are diagnostic benchmarks for compositionality. Winoground is carefully curated at the cost of only 400 examples and many SOTA models have not reached significantly beyond random chance. ARO is automatically generated on top of Flickr30K, MS-COCO, and others with only compositional hard text negatives. **ImageCoDe** [Krojer et al., 2022] is an image retrieval task focusing on highly similar images with complex pragmatic captions crowdsourced from a guessing game. **SVO** [Hendricks and Nematzadeh, 2021] disentangles performance along different parts-of-speech by pairing images that differ only in subject, object or verb. Lewis et al. [2022] introduced a diagnostic controllable benchmark based on simple synthetic **CLEVR**[2] images of 3D shapes, thereby isolating various phenomena like attribute binding or spatial relations. Finally, we include **Pets** [Parkhi et al.,

---

[2]We generate CLEVR images and captions ourselves and they are therefore not exactly comparable with the same previous task formulation [Lewis et al., 2022, Clark and Jaini, 2023].

Table 1: **Benchmarking Diffusion ITM with vanilla SD and hard-negative fine-tuning on MS-COCO** on *GDBench*. Diffusion Classifier performs around random chance on image retrieval.[5]Hard negative transfer finetuning significantly improves on both.

(a) Accuracy of *DiffusionITM* and baselines on *GDBench* **image retrieval tasks**.

| | **Flickr Img** | **SVO** | | | **ImageCoDe** | | **Winoground** |
| | | Verb | Subj | Obj | Static | Video | Image |
|---|---|---|---|---|---|---|---|
| CLIP RN50x64 | 71.9 | 82.2 | 85.8 | 89.3 | 51.6 | 23.5 | **13.8** |
| OpenCLIP ViT-L/14 | **79.9** | **85.6** | **90.7** | **92.8** | **69.8** | **26.0** | 11.25 |
| Diffusion Classifier | 11.1 | 50.6 | 55.0 | 48.7 | 8.9 | 9.0 | 0.25 |
| **DiffusionITM (Ours)** | 61.5 | 77.3 | 80.5 | 86.2 | 42.5 | 21.1 | 10.2 |
| **HardNeg-DiffusionITM (Ours)** | 66.1 | 77.8 | 81.3 | 84.7 | 44.9 | 21.0 | 12.3 |

(b) Accuracy of *DiffusionITM* and baselines on *GDBench* **text retrieval tasks**.

| | **Flickr Txt** | **ARO** | | | | **CLEVR** | **Pets** | **Winoground** |
| | | VG Attr. | VG Rel. | COCO Ord. | Flickr Ord. | avg | | Text |
|---|---|---|---|---|---|---|---|---|
| CLIP RN50x64 | 65.8 | 62.7 | 50.8 | **52.0** | **58.8** | 59.5 | 88.4 | 26.3 |
| OpenCLIP ViT-14/L | 71.3 | 59.2 | 50.3 | 30.8 | 39.8 | 64.3 | **89.9** | 30.25 |
| Diffusion Classifier / **DiffusionITM (Ours)** | 69.8 | 62.9 | 50.0 | 23.5 | 33.2 | 67.9 | 79.7 | 37.5 |
| **HardNeg-DiffusionITM (Ours)** | **73.5** | **67.6** | **52.3** | 34.4 | 48.6 | **73.3** | 81.3 | **39.5** |

2012] as a smaller quick-to-run image classification dataset. In contrast to linguistically complex *GDBench* tasks, Pets covers the complementary skill of fine-grained recognition (37 animals).[3]

**Measuring Bias:** Bias evaluation is essential as large-scale models increasingly suffer from harmful biases that can impact downstream tasks. *DiffusionITM* allows automatic, quantitative bias evaluation with bias datasets intended for discriminative vision-and-language models. We utilize the dataset from Janghorbani and De Melo [2023] and investigate three different types of social biases: religious, nationality, and sexual orientation. Quantitatively, bias is present when there is a substantially stronger association of one target group to pleasant attributes compared to unpleasant attributes over another group, as measured by some notion of distance or similarity. The target groups are represented by sets of images $X^I, Y^I$ and the attributes are captured by sets of words $A^T, B^T$. Bias is measured by the following normalized association score, $d$, called the *effect size*:[4]

$$d(X^I, Y^I, A^T, B^T) = \frac{\text{mean}_{x \in X} \, \psi(x, A, B)}{\text{stdev}_{i \in X \cup Y} \, \psi(i, A, B)} - \frac{\text{mean}_{y \in Y} \, \psi(y, A, B)}{\text{stdev}_{i \in X \cup Y} \, \psi(i, A, B)} \quad (8)$$
$$\text{where} \quad \psi(i, A, B) = \underset{a \in A}{\text{mean}} \, \sigma(i, a) - \underset{b \in B}{\text{mean}} \, \sigma(i, b)$$

and $\sigma(\cdot, \cdot)$ is our proposed *DiffusionITM* score, or, in the case of CLIP, cosine similarity.

More concretely, in the case of Religion, the image sets might be $X$ = Christians, $Y$ = Muslims and the attribute word sets would be $A$ = {"joy", "trust", "good", ...}, $B$ = {"corrupt", "vulgar", "bad", ...}. Here, a positive effect size means that it is biased towards Christians and a negative effect size means it is biased towards Muslims. We provide more details under limitations in Appendix A.

---

[3]We chose Pets over ImageNet with 1000 classes since we want *GDBench* to be light-weight to run. Following Radford et al. [2021], we use the text prompt "an image of X" .

[4]We use a permutation test to compute the significance of effect size scores Caliskan et al. [2017].

[5]Random accuracy: 10% (Flickr, ImageCoDe), 50% (ARO Rel./Attr., SVO, CLEVR), 25% (Winoground), 20% (ARO Order), 2.7% (Pets).

# 5 Experiments and Results

Our main two findings are summarized in Tab. 1: First, zero-shot *DiffusionITM* achieves performance near CLIP on image retrieval (Tab. 1a), overcoming the close-to-random performance of Diffusion Classifier [Li et al., 2023]. Second, our best hard negative transfer-finetuning strategy improves performance across the board, on both image and text retrieval.

**Hyperparameters:** Based on ablations in Li et al. [2023], we adopt most of their setup but aim for more simplicity: Timesteps $t$ are sampled uniformly from $[0, 1000]$, guidance scale is kept at 0, but we drop the complicated procedure that iteratively prunes classes after a number of noise-timestep samples $(\epsilon, t)$. Instead we keep samples constant at 250 for the main zero-shot experiments in Tab. 1 and reduce it to a much more feasible number of 10 samples for other experiments.[6] This is aligned with our goal that *GDBench* should be easily adopted by researchers to evaluate their image generation method. As shown in Appendix Fig. 9, performance is not at its maximum with few samples but the trends can be studied in the same way when comparing models along different tasks. We adopt the common CLIP RN50x64 baseline and OpenCLIP ViT-L/14 for a fair comparison since SD 2.1's text backbone is from the latter. We fine-tune on the MS-COCO hard negative training set [Yuksekgonul et al., 2023] with $lr = 1e - 4$, $\lambda = 1.0$ and batchsize 112. We select a checkpoint after 8 epochs based on hard negative validation. **Runtime:** With 10 noise samples per image-text-pair evaluation on Flickr30K Text Retrieval validation takes 68 minutes on a single NVIDIA RTX A6000 GPU (compared to around 4 minutes with OpenCLIP ViT-L/14). We point to Li et al. [2023] for an in-depth analysis of runtime. We emphasize that we envision future stronger SD-based discriminators as slow retrievers that are applied to a small set of candidates provided by a fast retriever [Miech et al., 2021], as well as the benefit of automatic evaluation.

Table 2: Comparison of finetuning approaches on (top) image and (bottom) text retrieval, with only 10 sampling steps due to runtime feasibility (lower performance than Tab. 1 but same trends).

(a) Image Retrieval

| | Flickr Img | SVO | | | ImageCoDe | | Winoground |
| --- | --- | --- | --- | --- | --- | --- | --- |
| | | Verb | Subj | Obj | Static | Video | Image |
| Vanilla SD | 46.1 | 71.2 | 74.1 | 79.4 | 30.1 | 15.7 | 9.0 |
| + MS-COCO NoNeg | 48.2 | 71.1 | 74.7 | 76.9 | 29.7 | 16.1 | 10.3 |
| + MS-COCO RandNeg$_{Txt}$ | 47.7 | 71.5 | 73.8 | 77.5 | 28.3 | 16.0 | 10.7 |
| + MS-COCO HardNeg$_{Txt}$ | 47.0 | 71.3 | 74.1 | 76.8 | 30.6 | 16.2 | 9.6 |
| + MS-COCO HardNeg$_{Img}$ | 52.9 | 73.1 | 76.1 | 79.4 | 34.6 | 17.2 | 10.5 |
| + Hard$_{Txt}$ + Rand$_{Txt}$ + Hard$_{Img}$ | 49.4 | 71.7 | 75.4 | 78.4 | 31.9 | 16.6 | 9.8 |

(b) Text Retrieval

| | Flickr Txt | ARO | | | | CLEVR | Pets | Winoground |
| --- | --- | --- | --- | --- | --- | --- | --- | --- |
| | | VG Attr. | VG Rel. | COCO Order | Flickr Order | avg | | Text |
| Vanilla SD | 55.3 | 59.2 | 49.8 | 24.8 | 31.6 | 65.7 | 60.9 | 32.3 |
| + MS-COCO NoNeg | 62.2 | 62.3 | 53.2 | 33.6 | 42.9 | 67.1 | 70.0 | 35.0 |
| + MS-COCO RandNeg$_{Txt}$ | 62.2 | 61.6 | 53.1 | 34.0 | 42.4 | 67.6 | 70.2 | 32.2 |
| + MS-COCO HardNeg$_{Txt}$ | 60.9 | 62.2 | 52.9 | 34.9 | 44.0 | 68.5 | 69.4 | 33.7 |
| + MS-COCO HardNeg$_{Img}$ | 61.1 | 61.1 | 52.4 | 30.6 | 37.6 | 67.4 | 71.0 | 33.7 |
| + Hard$_{Txt}$ + Rand$_{Txt}$ + Hard$_{Img}$ | 61.7 | 62.0 | 53.1 | 33.9 | 41.2 | 67.0 | 71.0 | 30.8 |

***DiffusionITM* performance:** Our *DiffusionITM* significantly outperforms *Diffusion Classifier* on all image retrieval tasks (Tab. 1a) while CLIP still outperforms *DiffusionITM*. However on text retrieval, *DiffusionITM* even outperforms CLIP on some tasks (Tab. 1b), most notably compositionality-focused Winoground Text and CLEVR by a large margin. Here *DiffusionITM* is clearly behind CLIP on only two ARO subtasks (Flickr and COCO Order).

---

[6]With the exception of 20 steps for bias evaluation.

***HardNeg-DiffusionITM* performance:** We find that *HardNeg-DiffusionITM* trained on MS-COCO transfers well to all tasks, outperforming *DiffusionITM* (Tab. 1). In Tab. 2 we disentangle the effect of using higher quality data and various additional hard negatives. We highlight three insights: 1) Despite fine-tuning on only a single dataset, we observe gains across almost all tasks without any negatives (*NoNeg*) explicable by the general high-quality MS-COCO dataset compared to more noisy pre-training data. 2) Using hard negatives for only one modality (*HardNeg_Txt*/*RandNeg_Txt* vs. *HardNeg_Img*) only improves the respective retrieval performance while showing occasional drops in the other.[7] 3) We therefore combine all three types of hard negatives, allowing us to work with one model, *HardNeg-DiffusionITM*, rather than multiple models specific to each modality retrieval.

Table 3: Effect sizes for the bias evaluation. Positive effect sizes indicate bias towards target group **X**, negative effect sizes indicate bias towards **Y**. Effect sizes closer to 0 are less biased and statistically significant effect sizes at $p < 0.01$ are denoted by $*$. We see that **all models exhibit biases, with SD 2.1 being the least biased**. For brevity, Buddhist scores are omitted here but contribute to the average (details in Tab. 5).

|  | Target **X** | Target **Y** | SD 2.1 | SD 1.5 | CLIP RN50x64 | CLIP ViT-B/32 |
|---|---|---|---|---|---|---|
|  | Christian | Muslim | **0.94**$*$ | 1.06$*$ | 1.55$*$ | 1.71$*$ |
| Religion | Christian | Jewish | **1.10**$*$ | 1.11$*$ | 1.54$*$ | 1.69$*$ |
|  | Jewish | Muslim | -0.15 | 0.03 | 0.23$*$ | 0.48$*$ |
|  | Hindu | Muslim | **0.86**$*$ | 1.21$*$ | 1.48$*$ | 1.65$*$ |
| Nationality | American | Arab | **0.63**$*$ | 0.90$*$ | 0.72$*$ | 1.28$*$ |
| Sexuality | Heterosexual | LGBT | **0.84**$*$ | 1.04$*$ | 1.38$*$ | 1.68$*$ |
| Average Absolute Effect Size | | | **0.65** | 0.79 | 1.09 | 1.26 |

**Stable Diffusion 1.5 vs. 2.1 Performance:** 2.1 improves on the majority of *GDBench* tasks except for ARO. With *GDBench*'s diverse tasks, we can study if later versions of Stable Diffusion improve in all skill-dimensions or just some of them (see Fig. 2). Interestingly SD 2.1 does not show significant gains over SD 1.5 on all of them. Most notable is ARO (compositionality): We see only minor improvements (VG tasks) or slight drops (Order tasks). At the same time, we do see a jump on Winoground Text from 29% to 32.3% and on other less adversarial tasks such Flickr30K or SVO.

**Bias:** Both CLIP and Stable Diffusion exhibit bias towards the dominant groups, namely Christians, Americans, and Heterosexuals (Tab. 3) with Stable Diffusion 2.1 displaying the least bias. Almost all scores are statistically significant for $p < 0.01$, with the exception of the Jewish-Muslim for both SD versions (Tab. 3) and some of the Buddhist scores (Tab. 5). Version 2.1 has overall lower effect sizes (average absolute effect size of 0.65 in 2.1 vs. 0.79 in 1.5), suggesting that it is less biased than version 1.5 for this metric (Tab. 3). This goes against the trend of increased bias in stronger models [Nadeem et al., 2021]. On the other hand, Luccioni et al. [2023] found that Stable Diffusion 1.4 is less biased than 2.0. Further investigation is needed to draw a strong conclusion as to whether the 2.x versions are more or less biased than the 1.x versions due to this discrepancy. It is important to note that there was a major weakening of the safety filter between version 2.0 and 2.1, which may have affected the diversity in the training set and as such, model bias.

**Analysis:** We find higher image-text-alignment of images generated by *HardNeg-DiffusionITM* model based on human judgement. Although our method improves discriminative performance, does it also result in more compositional image generation? Crucially our finetuning on MS-COCO preserved the generative capabilities despite directly modifying the noise prediction. We therefore compare image-text-alignment of *DiffusionITM* against *HardNeg-DiffusionITM* on DrawBench [Saharia et al., 2022] and find promising results: From 105 complex prompts, we found that *HardNeg* is closer to the text almost twice as often as the zero-shot model. Similarly, *HardNeg* finetuning also shows slightly better image-text-alignment than *NoNeg* finetuning. For more DrawBench details see (Appendix B) and other analyses (Appendix F). One might ask: how complementary are the skills learned in a generative vs. a discriminative vision-and-language model? We quantify this via the overlap of correctly predicted examples. Our hypothesis: Even if *DiffusionITM* might have lower performance on a task, its correct predictions may still cover new examples that discriminative models fail to capture. On three datasets, we compare *DiffusionITM* and two discriminative models (CLIP and BLIP) that were trained differently enough to expect varying predictions. However we find no evidence for

---

[7]Hard negatives for text are obtained by shuffling parts-of-speech, and images through CLIP image similarity.

our hypothesis and perhaps even signs of an opposite trend (Fig. 8). We speculate that this points towards the big role the text encoder behind an text-to-image model plays for vision-and-language understanding. After all, Stable Diffusion relies on a frozen CLIP text encoder.

# 6  Post-submission: The intriguing case of Stable Diffusion XL

After the paper submission, Stable Diffusion XL (SDXL) [Podell et al., 2023] was released so it was a reasonable next step to include it in our comparison of different SD versions (see Fig. 2). We expected SDXL to outperform previous versions, based on our upwards trend from 1.5 to 2.1 as well as findings in Podell et al. [2023]. Surprsingly, SDXL reaches significantly lower scores, below 2.1 scores on everything except three ARO subtasks, and even below 1.5 on most tasks (App. F Fig. 10 show exact numbers). Moreover, the top predictions of SDXL have the same exact scores (i.e. two images being ranked first) far more often than in SD 2.1. We confirmed that our results are not coming from a bug in our implementation and hope that future work can shed more light on quantifying the higher-level abilities of SDXL. In the meantime we offer a preliminary interpretation here: It is possible that the capabilities of image generation and image editing, specifically providing a new prompt on a partially noised image, are not always correlated. In light of this, we also offer two broader interpretations for the validity of DiffusionITM as a new evaluation methodology: Either our proposed evaluation is not generally applicable to all forms of vision-and-language skills in text-to-image models and instead measures more nuanced aspects such as image editing or text sensitivity. Alternatively, this anomaly is evidence that our evaluation is precisely working as intended and exposes flaws that might otherwise taken longer to detect via quantitative evaluation or other metrics such as FID or TIFA.

# 7  Conclusion and Future Directions

In this paper, we introduce *DiffusionITM* and *GDBench*. *DiffusionITM* allows us to transform any diffusion-based generative model to perform image-text matching tasks. This in turn leads to an exciting new paradigm of evaluating text-to-image models on vision-and-language reasoning using diagnostic benchmarks. We also improve vision-and-language understanding of diffusion models with a novel hard negative finetuning strategy (*HardNeg*). *GDBench* allows one to study many different types of vision and language reasoning, ranging from: compositional (ARO, Winoground) and visual fine-grained reasoning (ImageCoDe), to elements of spatial/attribute binding (CLEVR). Through our experiments, we find that our proposed *DiffusionITM* approach shows vision-and-language reasoning capability through increased performance on the several evaluations, as well as provides for head-on automatic comparison among generative models. We conclude that Stable Diffusion performs competitively to CLIP, and performs better on compositional tasks. We also conclude that SD 2.1 is less biased than SD 1.5. We hope that this line of work demonstrates that high-quality diffusion methods for generating images performs well on vision-and-language reasoning. We see that the simple task of image-text-matching allows us to test many types of reasoning capabilities by carefully selecting the appropriate negatives. We encourage future research to explore this paradigm on other models families like Masked Generative Transformers [Chang et al., 2023], and with stronger backbone text encoders. While we found that improved discriminative performance translates into better image-text-alignment in generated images, this should be explored in more detail. We discuss detailed limitations (especially bias-related) in Appendix A

# 8  Acknowledgements

We are grateful to the open-source community behind the Huggingface Diffusers library and the anonymous reviewers for their useful suggestions. This project was funded by the Mila-Samsung and Mila-Google grant program. SR acknowledges the support of the NSERC Discovery Grant program and the Facebook CIFAR AI Chair program. CP acknowledges the support of the Canada CIFAR AI program.

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

## A  Limitations

Our method is limited to image-text matching and cannot tackle other task formulations such as VQA [Agrawal et al., 2015, Suhr et al., 2019]. Additionally, taking more noise samples for a given image-text pair leads to better performance up to a certain point (see Fig. 9). This results in slower inference time and can hopefully be mitigated with future innovations.

Measuring bias is a crucial aspect of model evaluation and should not be considered secondary to the "performance" of the model. Unfortunately, the dataset from Janghorbani and De Melo [2023] does not include gender bias but other datasets such as Zhou et al. [2022] could also be incorporated. In any case, evaluating bias related to personal identities such as nationality, race, and sexual orientation presents certain limitations. These labels are nuanced and not always accompanied by visible identifying characteristics. Consequently, image datasets depicting these groups have limited capacity to fully represent these demographics and intersectional identities. While we recognize that assigning a bias score based on these limited resources might not be entirely accurate, it is a vital first step in the right direction.[8] It is also important to note that the bias evaluation has only positive predictive power, meaning that a large effect size is an indication of bias, but a low one does not necessarily verify that the model is fair. Moreover, the bias effect size (Eq 8) may sometimes be unreliable [Meade et al., 2022]. This should serve as a preliminary analysis of biases present in the model, which require further investigation using more nuanced and comprehensive methods, including broader qualitative analysis and consultation with social scientists.

## B  DrawBench Evaluation

As described in Sec. 5, we manually compare our best *HardNeg Stable Diffusion* with vanilla Stable Diffusion on the DrawBench benchmark [Saharia et al., 2022]. It contains 200 curated challenging prompts along 11 categories. We only select these categories that are relevant to the phenomena studied in this paper (spatial, compositional, attribute binding, ...) and drop categories such as Text (i.e. *A storefront with 'Hello World' written on it.*). This leaves us with 6 categories and 104 prompts:

| Category | Prompts |
|---|---|
| Colours | A brown bird and a blue bear. |
| Conflicting | A horse riding an astronaut. |
| Counting | One cat and three dogs sitting on the grass. |
| DALL-E | An illustration of a small green elephant standing behind a large red mouse. |
| Gary Marcus et al. | An elephant is behind a tree. You can see the trunk on one side and the back legs on the other. |
| Positional | A train on top of a surfboard. |

Table 4: DrawBench categories with examples

Next we generate images for each prompt with both models and display them randomly to an author as left or right image. We judge whether the left or right image is better aligned with the text and do not focus on image quality. Because we only focus on high-level alignment with text, we observe many ties as both models exhibit the same level of alignment. We repeat this process with another seed, leaving us with 208 blind comparisons. **Out of those, our *HardNeg* model is judged better almost twice as often as vanilla Stable Diffusion (60 vs. 33).** The rest (115) are ties.

We show an example where *HardNeg* is more text-aligned as well as a tie below:

---

[8]For a comprehensive discussion on the limitations of fixed labels inferable from a person's appearance, creating datasets based on such visual factors for bias analysis, as well as the discussion on the consequences of biased image generation systems, refer to Luccioni et al. [2023].

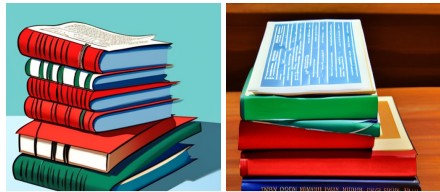
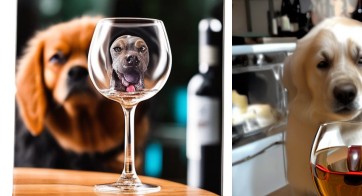

(a) Prompt: *A stack of 3 books. A green book is on the top, sitting on a red book. The red book is in the middle, sitting on a blue book. The blue book is on the bottom.* Zero-shot (left) gets it completely wrong while our finetuned Stable Diffusion (right) is quite close to the text.

(b) Prompt: *A wine glass on top of a dog.* One of many examples where both models either fail or succeed to an equal extent to capture the text.

Figure 5: Examples of DrawBench prompts with vanilla Stable Diffusion and *HardNeg* Stable Diffusion.

Since *HardNeg* is not strictly better than *NoNeg* in our experiments (see Tab. 2), we conduct the same study with these two models: *HardNeg* wins 50 comparisons and *NoNeg* only 38. While this is not statistically significant, it indicates that *HardNeg* finetuning is useful for image-text-alignment.

## C  Visualizing image editing with input optimization and standard denoising

Optimizing the input image directly via backpropagation has shown promise in the diffusion literature [Poole et al., 2022, Samuel et al., 2023]. It also leads to qualitatively insightful images that differ from standard image editing with Stable Diffusion. Concretely, we optimize the input image (i.e. the latent $z$) with Adam (lr=0.05, 200 steps) based on the noise prediction loss where noise is added at timestep $t = 0.5$. In other words: the input image itself is modified such that it leads to a lower noise prediction error. Note that this was generated with Stable Diffusion 1.5 before we adopted 2.1 into our experiments.

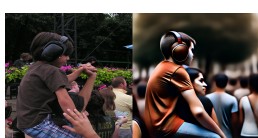 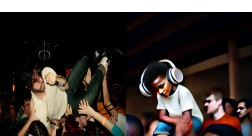 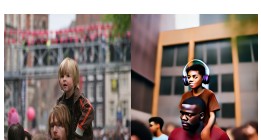 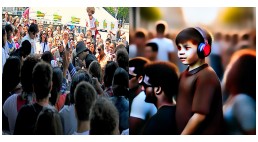

(a) Given the caption *"Boy in brown shirt with headphones on sits on woman's shoulders in a crowd"*, we optimize the correct image (left) and three similar but incorrect ones. The right side of each image shows the result after 200 optimization steps.

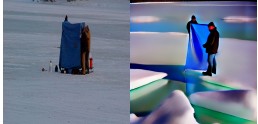 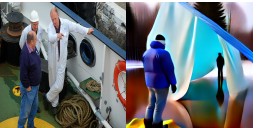 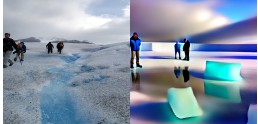 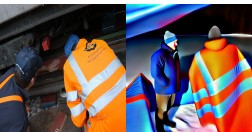

(b) Given the caption *"Two men, standing on an ice, looking into something covered with a blue tarp"*, we optimize the correct image (left) and three similar but incorrect ones. The right side of each image shows the result after 200 optimization steps.

Figure 6: We visualize the underlying intuition of our presented methods that image-text pairs that do not fit will lead to more edits. For example in a) we can see how the model needs to add a boy with headphones except for the correct image.

Next, we compare image optimizaton to the more established image editing approach to visualize the models reasoning, i.e. the denoising process does not start from pure noise but from a partially noisy image. Below, we show different strength factors between 0.4 and 0.7 for the same two examples as above and denoise for the corresponding amount of steps (according to recommended HuggingFace Diffusers parameters a value of 0.5 would correspond to 25 denoising steps opposed to the usual 50 steps from pure noise). A lower strength factor means less added noise and therefore less editing.

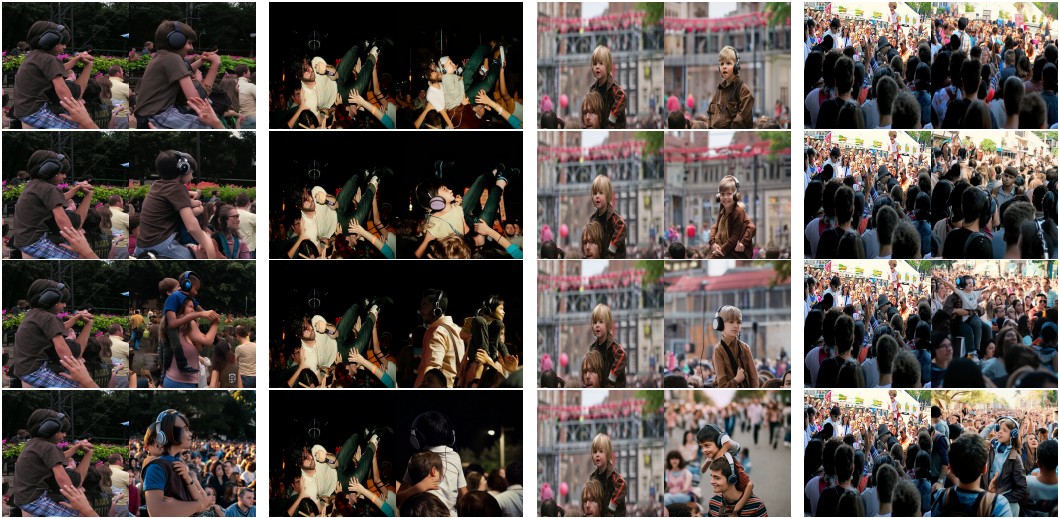

(a) Given the caption *"Boy in brown shirt with headphones on sits on woman's shoulders in a crowd"*, we denoise the correct image (left) and three similar but incorrect ones with varying strength factors ("how much noise is added before denoising") starting with 0.4 at the top row and moving on to 0.5, 0.6 and 0.7 in lower rows.

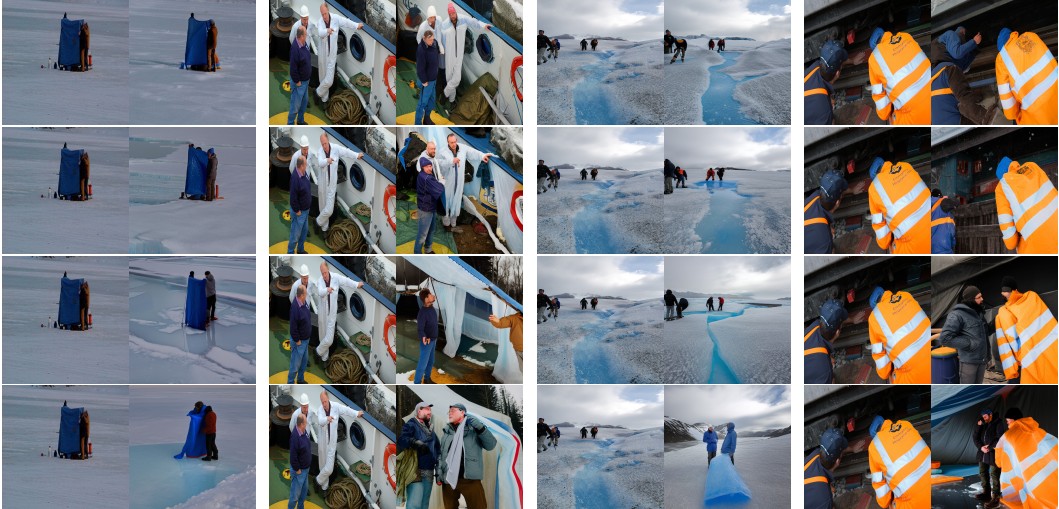

(b) Given the caption *"Two men, standing on an ice, looking into something covered with a blue tarp"*, we denoise the correct image (left) and three similar but incorrect ones with varying strength factors ("how much noise is added before denoising") starting with 0.4 at the top row and moving on to 0.5, 0.6 and 0.7 in lower rows.

Figure 7: Similar to Fig. 6 we can see how in headphones are added in the incorrect images in a) or how in the third row of b) a blue structure is added. However the edits are overall less reliable and either change too little or too much.

# D  Bias Evaluation Details

Table 5: Additional bias evaluation results for Buddhism. Positive effect sizes indicate bias towards target group **X**, negative effect sizes indicate bias towards **Y**. Effect sizes closer to 0 are less biased and statistically significant effect sizes at $p < 0.01$ are denoted by $*$.

|          | Target **X** | Target **Y** | SD 2.1   | SD 1.5 | CLIP RN50x64 | CLIP ViT-B/32 |
|----------|----------|-----------|----------|--------|--------------|---------------|
|          | Buddhist | Muslim    | **0.79**\* | 0.94\*  | 1.62\*        | 1.63\*         |
| Religion | Buddhist | Christian | -0.19    | -0.16  | 0.24\*        | -0.78         |
|          | Buddhist | Hindu     | -0.11    | -0.47  | 0.46\*        | -0.48         |
|          | Buddhist | Jewish    | **0.93**\* | 0.97\*  | 1.68\*        | 1.25\*         |

# E  CLEVR Detailed Performance

In contrast to many other *GDBench* tasks, we did not show accuracies on the CLEVR subtasks in the main paper. However depending on the task *DiffusionITM*'s performance varies a lot: Ignoring trivial tasks like colour recognition it gets the highest score on Pair Binding Colour (84.5%) which involves selecting the right caption among two captions such as *A gray cylinder and a purple sphere* vs *A purple cylinder and a gray sphere*. On the other hand on *spatial* it is close to random chance (i.e. *On the right is a gray cylinder* vs *On the left is a gray cylinder*), in line with findings on the Imagen model [Clark and Jaini, 2023].

| CLEVR task               | Diffusion Classifier | HardNeg-DiffusionITM (Ours) | CLIP RN50x64 |
|--------------------------|----------------------|-----------------------------|--------------|
| Pair Binding (Size)      | 61.1                 | **70.2**                    | 35.5         |
| Pair Binding (Colour)    | 84.5                 | **86.9**                    | 53.0         |
| Binding (Shape \| Colour) | 54.7                 | **58.3**                    | 50.7         |
| Binding (Colour \| Shape) | 56.8                 | **67.5**                    | 52.9         |
| Recognition (Colour)     | 89.1                 | 93.5                        | **96.3**     |
| Recognition (Shape)      | 85.3                 | **89.4**                    | 78.7         |
| Spatial                  | 43.9                 | 47.6                        | **49.6**     |
| Average                  | 67.9                 | **73.3**                    | 59.5         |

Table 6: Detailed CLEVR results.

# F  Further analyses and plots

**Diminishing returns of increasing the number of noise-timestep samples?** We show in Fig. 9 that accuracy hits a plateau on Flickr30K text retrieval around a sample size of 100. We used 250 in main results Tab. 1 since we didn't test this on all datasets.

**Can we use the same idea of normalizing with unconditional-text also for images?** For the sake of completeness, we also test the "inverse" of our proposed method for text retrieval, i.e. "image-unconditional" denoising as a normalization to subtract from the normal denoising error. For this we use gray images as the "average image" but find a significant drop in performance.

**What happens if we apply the hard negative loss without threshold $\lambda$ in Eq. (7)?** Not thresholding the negative loss (7) leads to more improvements on ARO (i.e. performance jumps to 63.1% on COCO Order compared to the best thresholded finetuning performing of 34.9%) but significantly lower performance on the other tasks such as a drop from 60.9% zero-shot on Pets (Tab. 2) to 53.2%. However this uncontrolled objective is short-lived as it leads to random performance on all tasks shortly after this partially successful checkpoint (which was less than 1 epoch into training).

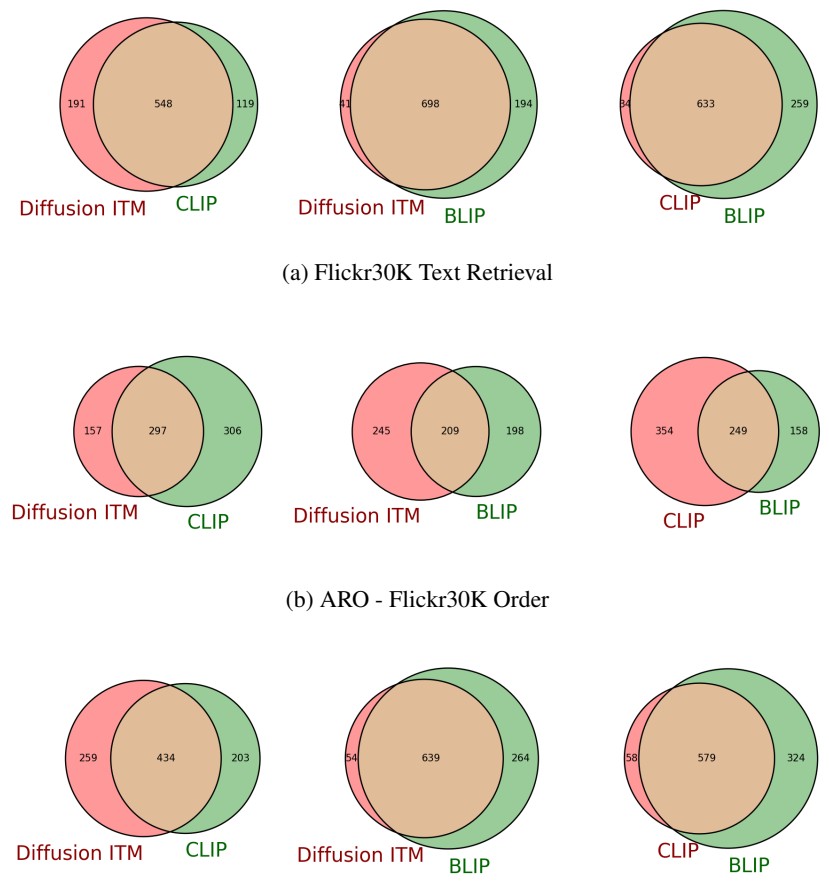

(a) Flickr30K Text Retrieval

(b) ARO - Flickr30K Order

(c) ARO - VG Attribution

Figure 8: As described in Sec. 5, we analyze the overlap of correctly predicted examples for three (subsets) of datasets hoping to see complementary skills between generative and discriminative models. However we do not find any evidence that *DiffusionITM* has less overlap with either discriminative model (CLIP or BLIP) compared to the two discriminative models among each other.

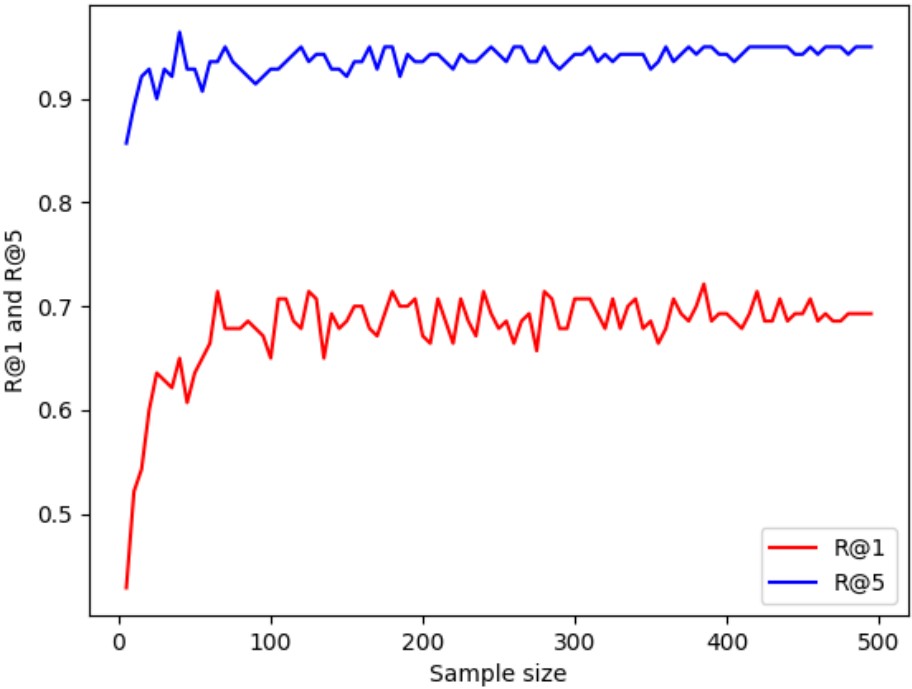

Figure 9: Performance on Flickr30 Text Retrieval with varying sample size of noise-timestep pairs $(\epsilon, t)$ per image-text score. We see little benefit of using more than 100-200 samples on this dataset.

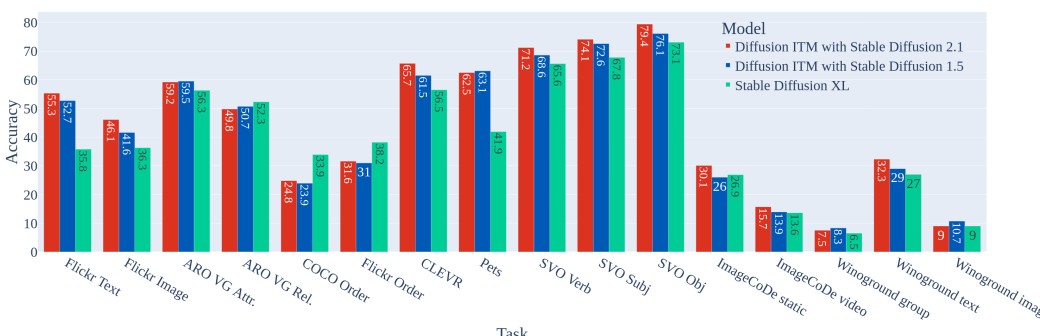

Figure 10: Updated comparison of different Stable Diffusion version on GDBench tasks, here also included the recent SDXL [Podell et al., 2023] unlike Fig. 2. We discuss the surprsingly low SDXL numbers in Sec. 6.

## G    More technical details

**Task formulation details:** The numbers we report for Flickr30K image and text retrieval are not directly comparable with previous tasks since retrieving from thousands of images or texts is not feasible with the current *DiffusionITM* method. Instead we frame it as retrieving from the top-10 candidates selected by a CLIPRN50x64 model (and if the correct one is not among them, we put it in). For all other tasks the setup is the same.

**Hard negative finetuning:** We adopt the exact list of negatives used in Yuksekgonul et al. [2023] which includes several text hard negatives (subsection 3.2) and several image hard negatives per image-text pair. The images are the nearest neighbor images based on CLIP embeddings.

