# OpenReview forum: "Are Diffusion Models Vision-And-Language Reasoners?"
_NeurIPS.cc/2023/Conference — NeurIPS 2023 poster_

### Official Review · Reviewer_ttjT · 2023-07-06

**Soundness:** 3 good
**Presentation:** 3 good
**Contribution:** 2 fair
**Rating:** 4
**Confidence:** 4

**Summary:**

The paper introduces Diffusion-ITM, a new method that directly adapts diffusion-based models to image-text matching tasks without retraining. Additionally, the authors collected a new benchmark called Generative-Discriminative Evaluation Benchmark (GDBench), which includes seven complex vision-and-language tasks and bias evaluation. The results show that Diffusion-ITM performs competitively on multiple tasks and surpasses CLIP. This paper underscores the significance of jointly considering discriminative and generative models and provides a new benchmark for future work.

**Strengths:**

* The paper presents a new method to adapt diffusion-based models for image-text matching tasks without retraining. This innovation has practical value to apply. The finding of the relative difference between the with and without text conditions (Fig. 3) is pretty interesting.
* The authors thoroughly evaluate the proposed method on various vision-and-language tasks, offering insights into its performance and potential biases.
* The introduction of new benchmark for image generation models over the discriminative tasks provides a useful tool for the research community and enables comparative analysis.
* This paper is well written and organized with clear logic to read.

**Weaknesses:**

Weaknesses:
* The Stable Diffusion includes a pre-trained CLIP text encoder and has itself been pre-trained on numerous image-text pairs. Therefore, Stable Diffusion has sufficiently learned the joint distribution of images and texts. The new finding is interesting but not surprising.
* The performance of the proposed method on more challenging benchmarks, like Winogroud, is relatively poor, suggesting room for potential improvement.
* The paper lacks theoretical analysis—since diffusion models are generative, applying them to a discriminative task would benefit from more theoretical insight.
* This paper ignores time cost/efficiency analysis. The Stable Diffusion inference is slow. According to Eq. 5, how to assign the t in your method? How many steps are needed and why?
* Minor issues, such as missing punctuation after equations, formatting errors, and a redundant space at the start of line 141, need to be addressed.


Question:
* Can the Stable Diffusion method be extended to more challenging image-text tasks like Visual Question Answering (VQA)?

**Questions:**

Please see weakness

**Limitations:**

Please see weakness

---

> ### Author Rebuttal · Authors · 2023-08-10
>
> We thank the reviewer for taking the time and effort to provide their detailed feedback on our submission! We are happy to note all the positive comments from the reviewer including:
> - “... finding of the relative difference between with and without text conditions (Fig. 3) is pretty interesting."
> - "... provides a useful tool for the research community and enables comparative analysis."
> - "well written and organized with clear logic to read"
> Similarly, other reviewers noted that:
> - “hard negative finetuning method is intuitive yet highly useful”
> - “Extensive ablations and experiments on variants of HardNeg, comparison between CLIP, BLIP, and Diffusion ITM [...]”
> - “paper studied bias [...] which should draw more attention to the community”
> - “Tackling a challenging problem, i.e., efficient quantitative evaluation of image generation models, with a simple yet effective method [...]”
>
> We believe the primary criticism can be attributed to our paper not clearly emphasizing a few key contributions. We hope to clarify these contributions and address the reviewer's concerns below.
>
> > "[...] The new finding is interesting but not surprising."
>
> We respectfully disagree with the reviewer that the result is unsurprising. In fact, **generative models typically have poor discriminative performance**; we point the reviewer to several prior works [4,5,6] that analyze this trade-off between generative and discriminative capabilities.
>
> Our contribution has been to analyze the discriminative capability of a new class of generative models i.e. denoising diffusion models. Little work on this topic exists prior to our submission. We admit that considering their powerful capabilities, it may be anticipated that they might perform well on discriminative tasks. Nevertheless, we fail to see how an “interesting but unsurprising” result is considered a weakness, especially in light of our extensive experimental evaluation.
> Going into the project we had two plausible hypotheses. 1) The generative objective might lead to deeper understanding of composition of visual scenes beyond the frozen text-encoder which has been shown to mostly work as a BOW model. Or 2): The generative objective might be focused on low-level details at the expense of leveraging the semantics from the text-encoder (i.e. unlearning semantics). **Both hypotheses needed to be empirically tested - hence our paper!**
>
> > "The performance [...] on more challenging benchmarks like Winoground [...] suggests room for potential improvement."
>
> We would like to emphasize that **progress on Winoground has been generally very slow** ([Why is Winoground Hard? Investigating Failures in Visuolinguistic Compositionality](https://aclanthology.org/2022.emnlp-main.143)) despite several papers and research groups having tried to tackle it, and similarly with benchmarks like ARO. We believe it is unfair to see this as a weakness of our specific paper and not the nature of well-designed hard tasks.
>
> > "[...] since diffusion models are generative, applying them to a discriminative task would benefit from more theoretical insight."
>
> We provide theoretical insight in the paragraph on Eq. 4 and 5. These are specific justifications for our method and not about the general concept of making a generative model into a discriminative one. We plan to expand the Related Work subsection on "Repurposing Text-Conditioned Diffusion Models", i.e. discussing how several families of generative models are adapted differently to a discriminative setting . We already have an older draft where we cite work such as [1,2,3]. Is this what you had in mind?
>
> > This paper ignores time cost/efficiency analysis. The Stable Diffusion inference is slow. According to Eq. 5, how to assign the t in your method? How many steps are needed and why?
>
> We provide time cost/efficiency analysis in the Appendix (see Fig. 9). Based on feedback from other reviewers as well, this will be **moved more prominently to the main paper**.
> Both concurrent works study time cost/efficiency analysis thoroughly, especially [Li et al. 2023], and our paper focused on additional contributions.
> Regarding choosing how many timesteps t to sample, we specify in Section 5 "Experiments and Results" under Hyperparameters, that we choose sample_size=250 for the main experiments and sample_size=10 for further ablations (sampled uniformly from [0,1000]).
>
> > Minor issues (punctuation, formatting) [...]
>
> Thank you for pointing these out! We will fix these.
>
> > Can the Stable Diffusion method be extended to [...] VQA?
>
> There is no elegant way but we did consider it (see Limitations section).
> Prior works have tried creating a "caption" by concatenating question+answer and then using old-school VQA evaluation (treating it as classification).
> However this is **impractical, and we discard it for two reasons**:
> First, our main goal was to evaluate SD off-the-shelf, or with minimal changes to its objective. If you treat SD as a backbone you can add all sorts of architectures on top. It is common to use the middle U-Net layer as the image-text representation, see concurrent work (Li et al., 2023).
> Second, VQA comes with its own problems. While it is “the” standard VL task, we believe that more **recent, often ITM-based, diagnostic benchmarks are more suitable to target phenomena such as compositionality**.
>
> We hope we have sufficiently answered all of your comments point by point, and are happy to engage further on more questions! Would you consider increasing your ratings given the clarifications?
>
> (Due to character limit constraints we provide the 6 citations from the main text as a comment after the rebuttal deadline)

---

> > ### Author Response · Authors · 2023-08-10
> > **Added Citations**
> >
> > [1]: Flow-GAN: Combining Maximum Likelihood and Adversarial Learning in Generative Models (Grover et al., AAAI, 2018)
> >
> > [2] Pixel recurrent neural networks (Van Den Oord et al., ICML 2016)
> >
> > [3]: [Diffusion Models as Masked Autoencoders](https://arxiv.org/abs/2304.03283) (Wei et al, Arxiv 2023)
> >
> > [4] On discriminative vs. generative classifiers: A comparison of logistic regression and naive bayes. (Ng and Jordan, NIPS 2001)
> >
> > [5] On the generative-discriminative tradeoff approach: Interpretation, asymptotic efficiency and classification performance. (Xue and Titterington, Computational Statistics & Data Analysis, 2010)
> >
> > [6] Training Normalizing Flows with the Information Bottleneck for Competitive Generative Classification. (Ardizzone et al., NeurIPS 2020)

---

> > > ### Comment · Reviewer_ttjT · 2023-08-14
> > > **After Rebuttal**
> > >
> > > Thanks to the detailed response from authors. I agreed with the answer to point one where the two hypotheses are essential. But I did not  draw the confident conclusion from the paper to support the claims of "deeper understanding of composition of visual scenes" and "focused on low-level details at the expense of leveraging the semantics from the text-encoder". How to prove this according to the experimental results of the paper?

---

> > > > ### Author Response · Authors · 2023-08-15
> > > > **Additional explanation of hypotheses**
> > > >
> > > > Thank you for your response and willingness to discuss our paper further!
> > > >
> > > > > not draw the confident conclusion from the paper to support the claims of "deeper understanding of composition of visual scenes" and "focused on low-level details at the expense of leveraging the semantics from the text-encoder"
> > > >
> > > > Briefly put, if Stable Diffusion had outperformed CLIP across the board, we could have concluded that the generative objective adds "something" that CLIP alone does not capture, and we illustrated that point with the broader phrase "deeper understanding of composition of visual scenes" since some of our benchmarks explicits focus on this type of skill.
> > > >
> > > > On the other hand, if CLIP had outperformed Stable Diffusion across the board, we could have concluded that somehow the generative objective "destroys" something in the CLIP text encoder (or that the CLIP visual encoder plays a major) role. With "focused on low-level details at the expense of leveraging the semantics from the text-encoder" we simply tried to convey a convincing explanation for such a behavior: The majority of the loss might be used towards aesthetics of the image (realistic texture etc) and not complex image-text alignment.
> > > >
> > > > **The reality in our paper is more nuanced**: Stable Diffusion performs quite well on some tasks (CLEVR and Winoground) while performing lower on others (Flickr30K, ImageCoDe,...). So one could draw the conclusion that the **generative objective helps with skills like spatial reasoning/attribute binding (CLEVR) and some challenging compositional examples (Winoground)** but not so much for other tasks like generic image/text retrieval (Flickr30K).
> > > >
> > > > To be more concrete, regarding “deeper understanding of composition of visual scenes”: Winoground, ARO and CLEVR benchmarks are particularly targeting this deeper understanding. Here DiffusionITM reaches 37.5% on Winoground (CLIP 26.3%), significantly lower numbers on ARO subsets such as 62.9% on VG Atrribution (CLIP 67.6%) and 67.9% on CLEVR (CLIP 59.5%).
> > > > This is **captured in Table 1 of the Rebuttal PDF** (which is an updated version of the paper Table 1 with a second baseline).
> > > > Even with an updated stronger CLIP baseline, as suggested by Reviewer JbKR, DiffusionITM still has the upper hand for Winoground and CLEVR.
> > > > For the other tasks (Flickr30K, SVO, ImageCoDe, Pets), while it is close on some, overall CLIP does a better job.
> > > > So we conclude that the generative objective can help but not always and can sometimes be detrimental.
> > > >
> > > > Since this is a major question of our paper, we can address it more directly in detail with the additional page for camera ready!
> > > >
> > > > Let us know if this answers your question. It is a long and hard research problem to answer the questions we raised in our paper (Are diffusion models vision-and-language reasoners?) but we believe it is a good start! It is not the final answer but hopefully inspires future work to investigate with other methods.

---

### Official Review · Reviewer_cFkA · 2023-07-08

**Soundness:** 2 fair
**Presentation:** 3 good
**Contribution:** 4 excellent
**Rating:** 6
**Confidence:** 3

**Summary:**

Recently diffusion-based text-to-image generation models have evolved rapidly, but it’s still challenging to evaluate them quantitatively in an efficient way.

This paper smartly converts evaluations of Stable Diffusion based image generation tasks as simpler image-text matching tasks (e.g. image-text retrieval), and proposes an image-text benchmark (GDBench) including 8 carefully-chosen tasks to measure generative model’s fine-grained properties (e.g. compositionality) and fairness. CLIP and Stable Diffusion models are evaluated on this new benchmark.

Overall, this paper bridges discriminative and generative model evaluations, which could be inspiring to the image generation community and improve the quantitative evaluations of generative models.


**Strengths:**

Tackling a challenging problem, i.e., efficient quantitative evaluation of image generation models, with a simple yet effective method (predicting the noise of the diffusion for image-text alignment).

Proposing an image-text benchmark (GDBench), covering diverse dimensions for measuring image generation (e.g. semantics, compositionality, fairness), which is beneficial for the image generation community.

Well written and easy to read.


**Weaknesses:**

The biggest concern on my side is the lack of enough evidence to show such a ITM eval aligns well with SD model quality, which IMO can’t be strongly supported by the evals to compare CLIP and SD models on retrieval numbers. If possible, I would suggest at least to prepare a few SD models with different capability (e.g. trained with different numbers of steps), and show they have different numbers on GDBench; bonus: give some qualitative examples for these models conditioned on the same text.

**Questions:**

As in “Weaknesses”: how to prove the proposed method can measure the capability of SD models?

To measure discriminative vision-language models, both classification and retrieval tasks are usually used. Besides “pets”, could we also add some other common 0-shot classification tasks as well? E.g., INET (most common) or ObjectNet (to measure robustness).

minor question/suggestion: this paper’s citation seems to use a different format (e.g. “[name year]”) than others (e.g. “[number]”)? The latter is more convenient to read, especially for prints.


**Limitations:**

This paper has addressed model bias in one of GDBench tasks.

---

> ### Author Rebuttal · Authors · 2023-08-10
>
> Dear Reviewer,
>
> Thank you for your kind and insightful comments! We are thrilled by your comment that the paper is "Well written and easy to read".
>
> We are glad you recognized that we are **"tackling a challenging problem, i.e., efficient quantitative evaluation of image generation models"**, with "diverse dimensions [...] ( e.g. semantics, compositionality, fairness)".
>
> Similarly, other reviewers noted that:
> - "[...] This innovation has practical value to apply. The finding of the relative difference between the with and without text conditions (Fig. 3) is pretty interesting."
> - "new benchmark for image generation models over the discriminative tasks provides a useful tool for the research community and enables comparative analysis."
> - “hard negative finetuning method in particular is intuitive yet highly useful”
> - “Extensive ablations and experiments on variants of HardNeg, comparison between CLIP, BLIP, and Diffusion ITM, and the relationship between the number of timesteps and the performance”
> - “Paper studied bias in state-of-the-models, which should draw more attention to the community”
>
> We address concerns and suggestions below, highlighting the contributions of our paper and describing additional experiments we have conducted.
>
> > "[...] prepare a few SD models with different capability (e.g. trained with different numbers of steps), and show they have different numbers on GDBench; bonus: give some qualitative examples for these models conditioned on the same text"
>
> We liked your main suggestion and have tested one more model, specifically the recently introduced [Stable Diffusion XL](https://arxiv.org/pdf/2307.01952.pdf). SD-XL comes with new hyperparameters and pre-processing steps and we are therefore in the process of confirming our results and some interesting findings. This is why we decided to not present rushed preliminary numbers here in the PDF.
> We also plan to include one more model for camera ready, i.e. SD-XL 0.9 or DeepFloyd (with a T5 text encoder), and as you suggested, are currently conducting more analysis similar to the DrawBench study already in the appendix. Note that these models were released after our main submission. Regarding “qualitative examples”, we provided additional analysis of HardNeg vs NoNeg setups in the new PDF (see global response).
>
> > “add some other common 0-shot classification tasks as well? E.g., INET (most common) or ObjectNet (to measure robustness).”
>
> Our goal and **contribution is primarily to study more complex vision-and-language reasoning** which is why we did not include ImageNet containing simple single-object images. These tasks were also studied in detail in related work like (Li 2023)[https://arxiv.org/abs/2303.16203]) and to our knowledge most recent models of vision-and-language also focus less on object recognition.
> That being said, GDBench doesn’t have to be static and ObjectNet looks like a good fit to test deeper understanding! For camera ready, we might either replace Pets with it or add it.
>
> Regarding citation format, thanks for the suggestion, we will revise the format in the final version.
>
> Thank you again for your helpful review! We believe we have addressed all of your concerns point by point. Given this, would you consider increasing your rating of our paper?

---

> > ### Comment · Reviewer_cFkA · 2023-08-18
> >
> > > Our goal and contribution is primarily to study more complex vision-and-language reasoning
> >
> > Makes sense. There's one more dataset for VL reasoning which isn't mentioned in this paper: Visual Spatial Reasoning. Added there in case it's helpful.
> >
> > Thank the authors for the detailed explanation. I would like to raise my rating to 6 and look forward to seeing more SD models in the camera-ready version.

---

> > > ### Author Response · Authors · 2023-08-21
> > >
> > > Thank you for your response and trust in our paper!
> > > We also appreciate the pointer to VSR, it is definitely exactly the kind of task we were interested in. We are considering adding it but the only issue for now is that the task is not directly cast as image-text-matching. The authors mention that they do it for zero-shot CLIP via negating the sentence but we are not sure if this is the right approach since it tests for negation understanding at the same time. In any case, we will investigate this further and are grateful for the pointer!

---

### Official Review · Reviewer_T2Yx · 2023-07-09

**Soundness:** 3 good
**Presentation:** 3 good
**Contribution:** 3 good
**Rating:** 4
**Confidence:** 4

**Summary:**

This paper studies how to use a pre-trained text-to-image generative diffusion model to do discriminative tasks like image and text matching. Building upon previous works [Li et al. 2023, Clark and Jaini 2023], this paper introduces two technical contributions: 1) Unconditional normalization largely improves image retrieval; 2) Tuning the diffusion model with hard negatives on better-quality data like MSCOCO. The paper also introduces a benchmark to evaluate different diffusion models by image and text matching performance, and specifically studies the biases of two versions of Stable Diffusion model.

**Strengths:**

1. The two technical contributions (unconditional normalization & tuning on MS COCO) are effective and well ablated.

2. The introduced benchmark could be useful for the community to evaluate the prompt-following ability and biases new diffusion models.

3. The paper studied bias in state-of-the-models, which should draw more attention to the community.

**Weaknesses:**

1. The generative-for-discriminative methods typically take many feedforwards of the diffusion model to evaluate a single sample. Although the paper briefly mentions its slow runtime at the end of the Appendix, I think the computational cost is an important factor of the proposed algorithm and should be rigorously presented and studied in the main text, especially when the paper proposes a benchmark - A clear study on the computational cost would be important to others who plan to evaluate on this benchmark.

2. The GDBench covers several hard image-and-text matching datasets. However, I think results on other standard discriminative datasets would be helpful for the audience to understand the advantage of the proposed algorithm and compare to previous works. For example, how does HardNeg-DiffusionITM work on ImageNet compare to Diffusion Classifier [Li et al. 2023] / DiffusionITM (Ours) in Table 2(b)? I understand ImageNet is a 1000-way classification and would be a long time to run for this kind of methods (therefore studying runtime is important as mentioned above), but Diffusion Classifier [Li et al. 2023]  has already reported their performance on ImageNet and an apple-to-apple comparison would be very helpful. Same applied to the CLEVR dataset. Another question is how tuning on MSCOCO works for image-text matching on MSCOCO?

3. I am not sure what is the use case to introduce MSCOCO tuning into Diffusion Classifier. It indeed improves the performance of image and text matching compare to the plain one, but all these methods are too slow (or not good enough) to be practically applied. On the other hand, tuning the weights of diffusion models prevents it from serving as a tool for evaluating diffusion models. Then in what case do we need MSCOCO tuning?

**Questions:**

1. I am not sure how much MSCOCO HardNeg is clearly better than MSCOCO NoNeg - To me, MSCOCO NoNeg seems to be even better than HardNeg quantitatively, or at least on par with in Table 2. How do the authors land with HardNeg as the default setting?

2. Why is Winground's random accuracy (25%) higher than the algorithms' accuracy in Table 1(a)?


**Limitations:**

The authors have adequately addressed the limitations and potential negative societal impact of their work.

---

> ### Author Rebuttal · Authors · 2023-08-10
>
> Thank you for your detailed comments and effort, showing that you engaged with our work.
> We are glad you found our “two technical contributions (unconditional normalization & tuning on MS COCO) effective and well ablated" and that our study contributes to drawing attention to bias.
>
> Similarly, other reviewers noted that:
> - “Tackling a challenging problem, i.e., efficient quantitative evaluation of image generation, with a simple yet effective method [...]”
> - "well written and organized with clear logic to read"
> - “hard negative finetuning method in particular is intuitive yet highly useful”
> - “Extensive ablations and experiments on variants of HardNeg, comparison between CLIP, BLIP, and Diffusion ITM …”
>
>  We believe the primary criticism can be attributed to our paper not clearly emphasizing a few key contributions. We hope to clarify these contributions and address the reviewer's concerns below.
>
> > [...] I think the computational cost [...] should be rigorously presented and studied in the main text [...] - important to others who plan to evaluate on this benchmark.
>
> The computational cost has been mentioned in the Appendix. Given that other reviewers were also looking for it, we will move the discussion of this important subject from the Appendix to the main paper and expand the analysis, as well as cite findings from other papers more prominently:
> Both concurrent works study time cost/efficiency analysis thoroughly, especially [1], and our paper focused on additional contributions, benefiting from their time cost findings.
> Specifically, we will a) conduct the same analysis of Fig. 9 for more than just 1 dataset and b) have implemented speed-up suggestions from R1 over the last week (with small improvements of 10-20%).
>
> > [...] I think results on other standard discriminative datasets would be helpful for the audience to understand the advantage of the proposed algorithm and compare to previous works [...] Diffusion Classifier has already reported their performance on ImageNet and an apple-to-apple comparison would be very helpful.
>
> Our **goal and contribution is  primarily to study more complex vision-and-language reasoning** which is why we did not include ImageNet containing simple single-object images. These vision-focused tasks were already studied in detail in related work like [1] and to our knowledge most recent models of vision-and-language also focus less on object recognition.
> Fair comparison to previous work is important, which is why we include CLEVR (a complex reasoning task) and put in effort to generate images+text as close to [2] as possible, since [2] did not publish their dataset. Can you clarify what you mean with “"same applied to the CLEVR dataset"?
> However since a) our focus is VL reasoning and b) our zero-shot text retrieval setting should be identical to DiffusionClassifier (Li et al., 2023) for the ImageNet case we do not see a strong need to add ImageNet. We do see value in ObjectNet as proposed by R4 and will consider it for camera ready!
>
> > Another question is how tuning on MSCOCO works for image-text matching on MSCOCO?
>
> If we understand the question correctly: We finetuned on MSCOCO train set so we can still test on the validation set. However the transfer setting can indeed not be studied. But our intuition is that Flickr30K covers a very similar task and hence it is not a problem for GDBench.
>
> > I am not sure what is the use case to introduce MSCOCO tuning into Diffusion Classifier.
>
>  Regarding “all these methods are too slow (or not good enough) to be practically applied”): a) it is possible diffusion or DiffusionITM become faster soon (i.e. one might get good performance with less samples) , and b) it is **common practice to have a fast retriever to narrow down the best results** (i.e. top-10) and next apply a slower sophisticated retriever for the final selection (see [Thinking Fast and Slow: Efficient Text-to-Visual Retrieval with Transformers](https://arxiv.org/abs/2103.16553)). Moreover, our work introduces a **generally new idea for incorporating negative pairs into generative pre-training** - which we believe is independent of applying it as ITM or evaluation of off-the-shelf models.
>
> > [...] how much MSCOCO HardNeg is clearly better than MSCOCO NoNeg
>
> That is a valid point and we authors discussed this during the submission. HardNeg is better for all image retrieval tasks (except Winoground which is in both cases below random and a tiny dataset). On the other hand, for text retrieval, NoNeg is better except for Pets. We see promise with negatives, especially hard image negatives for image retrieval.
> We investigated further regarding generative performance (specifically image-text-alignment) and conducted the same study as with DrawBench zeroshot vs. HardNeg in Appendix B, but this time on HardNeg vs. NoNeg. We found that **HardNeg has higher alignment, winning 50 of the comparisons while NoNeg only won 38** (see Rebuttal PDF). Hence, we chose to say that MSCOCO HardNeg is indeed better than MSCOCO NoNeg.
>
> > Why is Winground's random accuracy (25%) higher than the algorithms' accuracy [...]?
>
> This is very common behaviour: Most models tested in the Winoground paper reach below random! An intuition: Many VL models behave similar to a bag-of-words model, i.e. ignoring word order. So imagine a model that treats $c_0$ and $c_1$ (both containing same words in different order) as the same caption for many examples.
> At the same it has a default prior for one of the images and prefers it with both captions.
> Such behavior on many examples would lead to a below random score.
>
> We hope we have sufficiently answered all of your comments point by point, and are happy to engage further on more questions! Given this, would you consider increasing your rating of our paper?
>
> [1]: Your Diffusion Model is Secretly a Zero-Shot Classifier (Li et al., Arxiv 2023)
> [2]: Text-to-image diffusion models are zero-shot classifiers (Clark & Jaini, ICLR MRL Workshop 2023)

---

### Official Review · Reviewer_jr2c · 2023-07-09

**Soundness:** 3 good
**Presentation:** 3 good
**Contribution:** 3 good
**Rating:** 5
**Confidence:** 4

**Summary:**

This paper studies the discriminative capabilities of diffusion models measured by image-text matching. A new matching score computation enables text-based image retrieval beyond simply text retrieval in existing works. A new benchmark, augmented from existing image-text benchmarks, is proposed for researchers to evaluate current diffusion models.


**Strengths:**

1. DiffusionITM enables image retrieval by making simple and minimal changes to existing text retrieval methods. The method generates good results on simpler datasets and performs better than random on challenging ones.
2. The proposed benchmark covers diverse aspects but is also optimized for lightweight testing. Differences in a few methods have been demonstrated with the benchmark.


**Weaknesses:**

1. Evaluation on Flickr with CLIP retrieved negatives. This process might introduce bias towards or against CLIP based image diffusion models. For example, in Table 1a, CLIPRN50x64 obtains an accuracy of 71.9. According to line 545, does it mean that 28.1% of the cases CLIP does not select the correct positive? In order to fix this, maybe an evaluation against the original model-free Flickr retrieval metric is needed, e.g. measuring if CLIPRN50x64 or CLIP based method has an advantage/disadvantage compared with other methods, measured on the new simplified set vs. the original set. Similarly, the CLIP-based HardNeg training might be exploiting the bias, besides learning useful discrimination skills.
2. It is preferred to compare different kinds of diffusion model variants in the proposed benchmark and show some insights there. For example, one might see an advantage of T5 based text encoder in grammar/text related metrics, or one could study pixel based diffusion vs. latent diffusion etc. This comparison is challenging in obtaining the models but can provide more value to the community.
3. The current method makes an assumption that DiffusionITM, based on eq. 2 and 5, is a great way of ITM and will be the case for a while. However, given the field advancing so rapidly, researchers might discover a better zero-shot matching method that could infer quickly and outperform CLIP consistently for example. Then, at that time, why would researchers not adopt the original retrieval tasks, e.g. on Flickr with all images, but adopt GDBench with 10 images?
4. It is claimed that the proposed tuning strategy keeps the generation capability. Examples are shown in Figure 5 and positive human evaluation is shown, but is it possible to evaluate existing standard metrics and see the difference? FID might not apply as the tuning happens on COCO. One might even see better generation quality with discriminative tuning applied.
5. How is GDBench different from an ensemble of image-text matching evaluation metrics, except that it is made light-weight, e.g. measuring 10 samples. It seems DiffusionITM is a general enough method that can be evaluated on theoretically any image-text matching tasks. From another perspective, is it possible to show correlation of the GDBench metrics with real compositional generation quality, e.g. measured by human evaluation.
6. Given the limited samples in some of the evaluation datasets, what are the standard deviation for different runs? From Figure 9, it looks noisy.


**Questions:**

Table 1 typo: Difussion.
And see the weaknesses section for questions.

**Limitations:**

yes

---

> ### Author Rebuttal · Authors · 2023-08-10
>
> Thank you for your detailed comments and effort, showing that you engaged with our work! We are happy to note some of your positive feedback such as our method working with “minimal changes” and how “light-weight GDBench covers diverse phenomena”.
> Similarly other reviewers noted:
> - “Tackling a challenging problem, i.e., efficient quantitative evaluation of image generation, with a simple yet effective method”
> - “ The finding of the relative difference between with and without text conditions is pretty interesting."
> - " provides a useful tool for the research community and enables comparative analysis."
> - "well written and organized with clear logic to read"
> - “hard negative finetuning method in particular is intuitive yet highly useful”
> - “Extensive ablations and experiments on variants of HardNeg, comparison between CLIP, BLIP, and Diffusion ITM, …”
>
> We believe the primary criticism can be attributed to our paper not clearly emphasizing a few key contributions. We hope to clarify these contributions and address the reviewer's concerns below.
> > “It is preferred to compare different kinds of diffusion model variants in the proposed benchmark and show some insights”
>
> Based on your and R4's feedback, we are indeed **including more models and have experimented with the recently introduced Stable Diffusion XL** over the rebuttal period.
> SD-XL comes with new hyperparameters & preprocessing and we are therefore still confirming our results and interesting findings. This is why we decided to not present rushed preliminary numbers here in the PDF but will do so in camera ready after further investigation.
> We are not aware of any **published** open-source model using T5 text encoders but that would be an interesting addition. The only model we are aware of is the recently introduced DeepFloyd IF (without paper).
> At the time of writing the paper both models had not been released and most competitive open-source models are variations of SD.
>
> > [...] Researchers might discover a better zero-shot matching method that could infer quickly and outperform CLIP consistently for example. [...] Why would researchers not adopt [...] Flickr with all images but adopt GDBench with 10 images?
>
> While we do hope GDBench will be used by researchers, we can't predict the future for several years (realistically benchmarks fade but the GDBench might inspire the direction of future ones). For now we predict diffusion-based methods will stick for a while and will not become as fast as CLIP whose speed comes from encoding text & image separately. Any generative-model-turned-discriminative by definition needs to encode both together (=slow). So such a model should be considered as a second-step slow retriever after a fast simpler retriever (CLIP) has narrowed down the selection (details in [Thinking Fast and Slow: Efficient Text-to-Visual Retrieval with Transformers](https://arxiv.org/abs/2103.16553)). Recent trends in VL evaluation point towards prioritizing few high-qualitative examples over many easy examples. Therefore **future researchers will get the most value from evaluating few harder examples**, while also saving time with a slower model.
> > Evaluation on Flickr with CLIP retrieved negatives [...] might introduce bias towards or against CLIP based models
>
> We acknowledge this is not properly explained in the current version of the paper. However, we think a better research question in this situation is: "If CLIP was a first-step fast retriever to narrow down the search, could SD as a slow retriever improve upon re-using CLIP for the second step as well?" This was in fact an implicit motivation of us but not clearly enough stated in the paper. We will add a paragraph to the paper with a disclaimer as well as an explanation how these numbers are still insightful for the stated reason. While other experiments took priority over the last week, we plan to study another way to retrieve hard negatives (i.e. BERT-embeddings) for camera ready.
> For the finetuning we followed prior established methods and believe this has negligible bias.
> >  Is it possible to evaluate existing standard metrics [for generation] and see the difference?
>
> Our main goal of this paper is to **study higher-level semantic skills of these models which are not adequately captured by older standard metrics**. We therefore focus on small-scale human evaluation to validate our claim of on-par or better generation (see DrawBench analysis in Appendix and Global Rebuttal here). FID score would be insightful but, as explained in the global rebuttal, we are focusing on other high-priority experiments in this short time frame.
> > How is GDBench different from an ensemble of image-text matching evaluation metrics?
>
> GDBench is not a fully new dataset but here are some contributions:
> 1. We did make an effort to a) generate CLEVR again (not public before), b) make Flickr30K feasible and c) assemble the bias dataset.
> 2. Easy setup with our repo
> 3. Selecting tasks based on coverage+diversity of skills as well as feasibility criteria and recent trends, so that other researchers will not have to re-think this (like GLUE in NLP).
> But it is not set in stone and based on suggestions we plan to also include ObjectNet.
> On top, we are including more models to show how GDBench performance correlates with image-text-alignment ratings.
>
> > What are the standard deviations[...]? Fig. 9 looks noisy.
>
> Fig. 9 was generated with less than 200 of examples which still illustrates the point but we will fix the figure caption to mention this.
> We did not have time during rebuttal to compute std on all data but studied Winoground (as the smallest dataset) with a small sample size of 10 noise-timestep samples per datapoint, i.e. where we expect high std. Result: $31.5$% $\pm 0.43$.
>
> We hope we have sufficiently answered all of the reviewer’s comments point by point, and are happy to engage further on more questions! Considering this, we hope the reviewer increases their rating of our paper.

---

### Official Review · Reviewer_JbKR · 2023-07-11

**Soundness:** 4 excellent
**Presentation:** 4 excellent
**Contribution:** 4 excellent
**Rating:** 8
**Confidence:** 5

**Summary:**

This paper proposes novel techniques for improving the performance of text-to-image diffusion models on zero-shot image-text matching tasks. They first propose subtracting the unconditional denoising error $\|\epsilon - \epsilon_\theta(x_t, t)\|_2^2$, which reduces the problem in image retrieval where one image is a priori far more likely under all captions. The also propose using a hard negative finetuning loss to increase the denoising error on hard negatives, which improves its discriminative (and generative) capabilities. Finally, they assemble  GDBench, a benchmark of 7 existing datasets that they use to quantify different aspects of text-image alignment performance. They find that their proposed model significantly improves on prior work that uses text-to-image diffusion models (Diffusion Classifier), and is competitive with CLIP ResNet50x64. They also benchmark model bias and find that Stable Diffusion 2.1 is less biased than Stable Diffusion 1.5 and CLIP RN50x64.

**Strengths:**

- The paper proposes 2 novel modifications that significantly improve performance over Diffusion Classifier (a prior work on using diffusion models for text-image matching tasks).
  - The hard negative finetuning method in particular is intuitive yet highly useful. It is also evaluated extremely thoroughly in its effect on both discriminative and generative behavior.
- Authors also gather a set of 8 image-text-matching tasks into a proposed GDBench benchmark for easily measuring and comparing the performance of different ITM methods.
- Authors do a thorough analysis of the bias of Diffusion ITM, across versions of Stable Diffusion, as well as in comparison to a few CLIP models.
- Extensive ablations and experiments on variants of HardNeg, comparison between CLIP, BLIP, and Diffusion ITM, and the relationship between the number of timesteps and the performance of the method.

**Weaknesses:**

- Computational: The proposed method runs in $O(n)$, where $n$ is the number of candidate images or text captions to choose from. This becomes highly impractical for real retrieval applications, compared to approaches like CLIP where fast approximate nearest neighbor solutions work extremely efficiently.
- Baselines: while the method does outperform previous work on using diffusion models (e.g., Diffusion Classifier), it doesn't compare against the strongest discriminative baselines, namely OpenCLIP Vit-H/14. This is fair since Stable Diffusion 2 actually uses this model as its text encoder. The paper currently only compares against CLIP ResNet50x64, which is quite weak in comparison.

**Questions:**

**Main questions/issues**:
- How well does OpenCLIP ViT-H/14 do on these ITM tasks? This is the most relevant discriminative baseline since SD 2 uses its text encoder.
- The Flickr30K benchmark is biased against the CLIP model used to produce it. The CLIP model is used to produce the top 10 candidates, and *the correct candidate is added if it's not among the top 10*. If the model being evaluated is CLIP, it has to choose between 9-10 hard negatives, whereas other models will receive a boost from a weak ensembling effect with CLIP. I'd suggest another way of creating the 10 candidates without correlation to any model output.
- Equation 6: is there a major typo here? This appears to still minimize $\mathcal L_{neg}$.
- L 246: Hard negative finetuning uses $\lambda=-1$? Why is it negative when it's only used for $clip(\mathcal L_{neg}, |\lambda \mathcal L_{pos}|)$? Does this mean that $\mathcal L_{neg}$ is clipped to the range $[-\mathcal L_{pos}, \mathcal L_{pos}]$? Something is wrong here.

**Minor questions/comments**:
- Table 1b: Where does the 37.5 score come from with Diffusion Classifier/DiffusionITM on Winoground text? The Diffusion Classifier paper reports 34.0.
- Page 6, footnote 2: This is not too important, since the prompt is fixed across methods, but Pets prompt template used in the CLIP paper is "a photo of a {}, a type of pet." This probably increases scores across the board for all methods.
- L240-241: "we drop the complicated procedure that iteratively prunes classes after a number of noise-timestep samples" -- Diffusion Classifier doesn't use its adaptive strategy for its ITM experiments either, so its evaluation strategy matches DIffusionITM here.
- Appendix G (runtime) mentions that "batch size has diminishing returns in terms of runtime after batchsize=4." However, based on the Diffusion Classifier github, using FlashAttention + FP16 + `torch.inference_mode()` + batchsize=32 achieves about 80 evals/s on an A6000. I'd suggest double checking your implementation. Computation speed should be an issue overall with this method, but not to this level.

**Limitations:**

Yes, authors address the main limitations of their paper and method.

---

> ### Author Rebuttal · Authors · 2023-08-10
>
> Dear Reviewer,
>
> We are thankful for your detailed and extremely thorough review that shows you have engaged with the work and are very confident with the subject area!
> Thank you as well for highlighting strengths of the paper such as:
> - "The hard negative finetuning method in particular is intuitive yet highly useful. It is also evaluated extremely thoroughly in its effect on both discriminative and generative behavior."
> - "do a thorough analysis of the bias of Diffusion ITM, across versions of Stable Diffusion, as well as in comparison to a few CLIP models."
> - "Extensive ablations and experiments on variants of HardNeg, comparison between CLIP, BLIP, and Diffusion ITM, and the relationship between the number of timesteps and the performance"
> Similarly other reviewers also noted:
> - “Tackling a challenging problem, i.e., efficient quantitative evaluation of image generation models, with a simple yet effective method [...]”
> - "[...] This innovation has practical value to apply. The finding of the relative difference between the with and without text conditions (Fig. 3) is pretty interesting."
> - "new benchmark for image generation models over the discriminative tasks provides a useful tool for the research community and enables comparative analysis."
> - "well written and organized with clear logic to read" (another reviewer: “well written and easy to read”)
>
> We appreciate your very detailed comments with suggestions that even went into implementation details! We will address them below:
>
> > "How well does OpenCLIP ViT-H/14 do on these ITM tasks?"
>
> We fixed this valid concern by **running the ViT-L/14 baseline on all tasks** and will include it in Table 1 of the paper (and also the Rebuttal PDF here). ViT-H/14 seemed to be temporarily unavailable the last few days for download so we opted for the closest options based on the [OpenCLIP repo](https://github.com/mlfoundations/open_clip). The short answer is: It does better than our original baseline, as also shown in DiffusionClassifier, and we will discuss this in the camera ready paper.
>
> > "The Flickr30K benchmark is biased against the CLIP model used to produce it"
>
> Thank you for raising this concern! We acknowledge this is not properly explained in the current version of the paper and is therefore misleading. A **better research question in this situation might be**: "If CLIP was a first-step fast retriever to narrow down the search, could SD as a slow retriever improve upon re-using CLIP for the second step as well?" (see [Thinking Fast and Slow: Efficient Text-to-Visual Retrieval with Transformers](https://arxiv.org/abs/2103.16553) for in-depth discussion) This was an implicit motivation of ours but not clearly enough stated in the paper. We will add a paragraph to the paper with a **disclaimer as well as an explanation how these numbers are still insightful for the stated reason**. While other experiments took priority over the last week, we plan to study another way to retrieve hard negatives via BERT-embedding NN search for camera ready.
>
> Thank you for the smaller fixes and comments!
> You are right that $\lambda$ should not be $-1$. We will fix this!
>
> >  "Where does the 37.5 score come from with Diffusion Classifier/DiffusionITM on Winoground text?"
>
> We evaluated with 250 samples of uniform timesteps and beyond that we are not sure how to explain the difference. There is a small standard deviation and after a quick search in the DiffusionClassifier paper we did not find how many samples DiffusionClassifier used for Winoground.
>
> Regarding your suggestions regarding speed, we already used fp16 and now we also tried FlashAttention with small speed gains. Thank you! For our final repository or follow-up work we will optimize speed further. We do not have an A6000 unfortunately. We ran the DiffusionClassifier repo on our side the last days (to check something with Stable Diffusion XL) and it seemed comparably slow to ours but hard to tell due to the adaptive strategy!
>
> We hope we have sufficiently answered all of your comments point by point, and are happy to engage further on more questions! Considering this, we hope you increase your rating of our paper, and champion it for publication at this conference.

---

> > ### Comment · Reviewer_JbKR · 2023-08-19
> >
> > - OpenCLIP ViT-H/14 -- I just tried downloading it via the OpenCLIP repo and it seems to be available now. Could you run this baseline?
> > - Glad my smaller comments (like $\lambda$ for the hard negative finetuning) were helpful!
> > - Could you report the mean and variance of the DiffusionITM/Diffusion Classifier Winoground text score evaluation with 5 random seeds? I'm curious how much the randomness in evaluation affects the score, especially since there are so few examples in Winoground. This might also be a good idea for the other benchmarks where there are fewer test examples.
> > - I brought up Diffusion Classifier's A6000 inference speed since your Appendix G mentioned using an A6000 for inference. I'm still curious about the runtime of your implementation. How long does it take to evaluate a single Winoground text score (4 image-caption pairs x 250 timesteps = 1000 total evaluations) with your implementation?
> >
> > Hopefully the authors can add these results if they have time. If not, I still really like this paper and have increased my score.

---

> > > ### Author Response · Authors · 2023-08-21
> > >
> > > - ViT-H/14 works now! Loading it a week ago threw an error. We will run it soon for all datasets.
> > > - In response to reviewer jr2c, we computed std for Winoground but with 10 timesteps (which means higher variance) for 5 seeds. This was our response: "We did not have time during rebuttal to compute std on all data but studied Winoground (as the smallest dataset) with a small sample size of 10 noise-timestep samples per datapoint, i.e. where we expect high std. Result: $31.45$ % $\pm 0.43$"
> > > - We just ran our code with batchsize=1, 250 timesteps and a single GPU for 4 pairs, as you suggested. It took around 71 seconds, so 18 seconds per pair. This would be faster with multi-GPU and slightly faster with larger batchsize.
> > >
> > > Thank you again for your time and trust in the paper!

---

### Author Rebuttal · Authors · 2023-08-10

Dear Reviewers and Area Chairs,

First we would like to thank all reviewers for writing detailed and thoughtful responses. It raised interesting discussions among the authors and will make it an overall stronger paper. In particular, we are grateful that Reviewer JbKR gave us a a score of 7 with high confidence and very detailed technical feedback that showed his expertise in this domain.

Reviewers pointed out the following strengths of our paper:
1. **Method contributions**: “hard negative finetuning method in particular is intuitive yet highly useful” (JbKR), “two technical contributions (unconditional normalization & tuning on MS COCO) are effective and well ablated” (T2Yx) and "finding of the relative difference between the with and without text conditions (Fig. 3) is pretty interesting" (ttjT)
2. **Analysis**: “evaluated extremely thoroughly in its effect on both discriminative and generative behavior" (JbKR) and “extensive ablations and experiments on variants of HardNeg, comparison between CLIP, BLIP, and Diffusion ITM, and the relationship between the number of timesteps and the performance” (JbKR)
3. **Evaluation contribution**: “Tackling a challenging problem, i.e., efficient quantitative evaluation of image generation models, with a simple yet effective method” (cFkA), “benchmark covers diverse aspects but is also optimized for lightweight testing” (jr2c), “paper studied bias [...], which should draw more attention to the community” (T2Yx)
4. **Presentation**: “Well written and easy to read” (cFkA) and "well written and organized with clear logic to read" (ttjT)

On top of responding to concerns, we also addressed common feedback from reviewers with further empirical investigations, summarized here:
1. Reviewer JbKR pointed out that an **additional stronger baseline**  (CLIP ViT-H/14) is needed. The results are shown in the Rebuttal PDF (Tab. 1). The baseline is stronger overall and we will address this in the camera ready version.
2. Reviewer T2Yx asked about why we highlighted the **finetuning setup *HardNeg* over the *NoNeg* setup as our strongest model**. While performance on our discriminative benchmark is very close, we did conduct an additional human judgement study where we found that HardNeg has stronger image-text-alignment when it comes to **generative** performance (see details in rebuttal response to T2Yx and sample images in Rebuttal PDF)
3. Both jr2c and cFkA suggested to **evaluate more models** for reasons such as a) insights on model variants (i.e. T5 text encoder) and b) evidence that GDBench correlates with generative performance. Most strong open-source models boild down to variants of Stable Diffusion and we therefore spent most of this week evaluating the recently introduced Stable Diffusion XL. SD-XL comes with new hyperparameters and pre-processing steps and we are therefore still confirming our results and some interesting findings. This is why we decided to not present rushed preliminary numbers here but will do so in camera ready after further investigation. We are not aware of any open-source model (that also comes with a paper) using T5 text encoders but that would be an interesting addition. The only model we are aware of is the recently introduced [DeepFloyd IF](https://github.com/deep-floyd/IF) (without paper). **At the time of writing the paper both models had not been released.**
4. We provide standard deviation for jr2c.

Due to the short time frame we had to prioritize the most insightful experiments and responded to other comments in the rebuttal responses. Here the commom threads were the computational/speed cost of our method (JbKR, T2Yx, ttjT) and including more datasets (cFkA, T2Yx):
1. **Computational cost**: We agree with reviewers that computational cost is an important issue, and had previously discussed it in the Appendix. For the final paper we will move the discussion from the Appendix to the main paper, expand the analysis, and cite findings from other papers more prominently:
Both concurrent works study time cost/efficiency analysis thoroughly, especially Li et al. [2023], and our paper focused on additional contributions, benefiting from their findings. Hence this is already covered in previous literature.
We also emphasized that models such as DiffusionITM are not intended to be a fast retriever such as CLIP but more a “second-step slow retriever” that comes into play once a fast retriever has narrowed down the options to the hardest candidates. This is common practice (see [Miech et al., 2021](https://arxiv.org/abs/2103.16553)).
2. **More datasets**: Two reviewers suggested more vision tasks. Our **goal and contribution is primarily to study more complex vision-and-language reasoning** which is why we did not include ImageNet containing simple single-object images. These vision-focused tasks were already studied in detail in related work like [Li 2023](https://arxiv.org/abs/2303.16203) and to our knowledge most recent models of vision-and-language also focus less on object recognition. Fair comparison (raised by T2Yx) to previous work is important, which is why we include CLEVR (a complex reasoning task) and put in effort to generate images+text as close to [2] as possible ([2] did not publish their dataset). We do see value in ObjectNet as proposed by cFkA and will consider it for camera ready.

We plan on studying the following for a potential camera ready version:
- bias in our modified Flickr30K task (i.e. how to choose hard negatives for Flickr30K)
- include ObjectNet as an additional task

As much as we would like to run all possible experiments, it is unclear whether we will have enough time to fulfill all other suggestions. However we will mention these points in the limitations section!

Overall, we believe we addressed the main criticisms and engaged faithfully and constructively with the feedback. We are looking forward to fruitful exchanges the next weeks to further improve the work!

---

### Decision · Program_Chairs · 2023-09-21

**Decision:**

Accept (poster)

**Comment:**

This paper proposes novel techniques for improving the performance of text-to-image diffusion models on zero-shot image-text matching tasks. The paper received mixed reviews.

On the positive side, the proposed method is novel, intuitive and interesting, achieves reasonably good results. The paper also introduced benchmark GDBench covers diverse skills and provides a useful tool for evaluating image-text alignment. The evaluation is comprehensive, which includes extensive ablations and the model bias study. The paper is well written and easy to follow.

On the negative side, the computational cost of the proposed method is high, making it impractical for real applications compared to efficient methods like CLIP. Performance on very challenging datasets like Winoground is still quite low. More comparisons to other diffusion model variants would provide more insights. Additional experiments validating that the benchmark correlates with generative quality would be useful. More theoretical analysis on repurposing generative models as discriminative models could provide insight.

Overall, this paper provides some interesting observations which may bring useful discussions to the community.